



# Multi-year simulations at kilometre scale with the Integrated Forecasting System coupled to FESOM2.5/NEMOv3.4

Thomas Rackow[1,2], Xabier Pedruzo-Bagazgoitia[1], Tobias Becker[1], Sebastian Milinski[1], Irina Sandu[1], Razvan Aguridan[1], Peter Bechtold[1], Sebastian Beyer[2], Jean Bidlot[1], Souhail Boussetta[1], Michail Diamantakis[1], Peter Dueben[1], Emanuel Dutra[3], Richard Forbes[1], Helge F. Goessling[2], Ioan Hadade[1], Jan Hegewald[4], Sarah Keeley[1], Lukas Kluft[5], Nikolay Koldunov[2], Alexei Koldunov[2], Tobias Kölling[5], Josh Kousal[1], Kristian Mogensen[1], Tiago Quintino[1], Inna Polichtchouk[1], Domokos Sármány[1], Dmitry Sidorenko[2], Jan Streffing[2], Birgit Sützl[1], Daisuke Takasuka[6,7], Steffen Tietsche[1], Mirco Valentini[1], Benoît Vannière[1], Nils Wedi[1], Lorenzo Zampieri[8], and Florian Ziemen[9]

[1]European Centre for Medium-Range Weather Forecasts (ECMWF), Bonn, Germany; Reading, UK
[2]Alfred Wegener Institute, Helmholtz Centre for Polar and Marine Research (AWI), Bremerhaven, Germany
[3]Portuguese Weather Service (IPMA), Lisbon, Portugal
[4]Gauß-IT-Zentrum, Braunschweig University of Technology (GITZ), Braunschweig, Germany
[5]Max-Planck-Institute for Meteorology (MPI-M), Hamburg, Germany
[6]Atmosphere and Ocean Research Institute (AORI), The University of Tokyo, Kashiwa, Japan
[7]Japan Agency for Marine-Earth Science and Technology (JAMSTEC), Yokohama, Japan
[8]Foundation Euro-Mediterranean Center on Climate Change (CMCC), Bologna, Italy
[9]Deutsches Klimarechenzentrum (DKRZ), Hamburg, Germany

*Correspondence to*: Thomas Rackow (thomas.rackow@ecmwf.int)

**Abstract.** We report on the first multi-year km-scale global coupled simulations using ECMWF's Integrated Forecasting System (IFS) coupled to both the NEMO and FESOM ocean-sea ice models, as part of the H2020 Next Generation Earth Modelling Systems (nextGEMS) project. We focus mainly on the two unprecedented IFS-FESOM coupled setups, with an atmospheric resolution of 2.8km and 4.4km, respectively, and the same spatially varying ocean resolution that reaches locally below 5km grid-spacing. This is enabled by a refactored ocean model code that allows for more efficient coupled simulations with IFS in a single-executable setup, employing hybrid parallelisation with MPI and OpenMP. A number of shortcomings in the original NWP-focussed model configurations were identified and mitigated over several cycles collaboratively by the modelling centres, academia, and the wider nextGEMS community. The main improvements are (i) better conservation properties of the coupled model system in terms of water and energy balance, which benefit also ECMWF's operational 9 km IFS-NEMO model, (ii) a realistic top-of-the-atmosphere (TOA) radiation balance throughout the year, (iii) improved intense precipitation characteristics, and (iv) eddy-resolving features in large parts of the mid- and high-latitude oceans (finer than 5km grid-spacing) to resolve mesoscale eddies and sea ice leads. New developments made at ECMWF for a better representation of snow and land use, including a dedicated scheme for urban areas, were also tested on multi-year timescales. We provide first examples of significant advances in the realism and thus opportunities of these km-scale simulations, such as a clear imprint of resolved Arctic sea ice leads on atmospheric temperature, impacts of km-scale urban areas on the diurnal temperature cycle in cities, and better propagation and symmetry characteristics of the Madden-Julian Oscillation.



## 1 Introduction

Current state-of-the-art climate models with typical spatial resolutions of 50-100km still rely heavily on parametrizations for under-resolved processes, such as deep convection, the effects of sub-grid orography and gravity waves in the atmosphere, or the effects of meso-scale eddies in the ocean. The emerging new generation of km-scale climate models can explicitly represent and combine several of these energy-redistributing small-scale processes and physical phenomena that were historically approximated or even neglected in coarse-resolution models (Palmer 2014). The advantage of km-scale models thus lies in their ability to more directly represent phenomena such as tropical cyclones (Judt et al. 2021) or the atmospheric response to small-scale features in the topography, e.g. mountains, orography gradients, lakes, urban areas, and cities. The distribution and intensity (and particularly the extremes) of precipitation (Judt and Rios-Berrios 2021), winds, and potentially also temperature will be different at improved spatial resolution. Importantly, features of deep convection start to be explicitly resolved at km-scale resolutions. This does not only improve the local representation of the diurnal cycle, convective organisation, and the propagation of convective storms (Prein et al., 2015; Satoh et al., 2019; Schär et al., 2020), but can also impact the large-scale circulation (Gao et al. 2023). Ultimately, the replacement of parametrizations by explicitly resolved atmospheric dynamics is also expected to help narrowing the still large uncertainty range of cloud-related feedbacks and thus climate sensitivity (Bony et al. 2015; Stevens et al. 2016).

Km-scale resolutions are also particularly beneficial for the ocean, where mesoscale ocean eddies (Frenger et al. 2013), leads opening up in the sea ice cover, and the response of oceanic heat transport to the presence of narrow canyons (Morrison et al. 2020) can be studied directly. The small scales in the ocean, in particular mesoscale ocean eddies, have large-scale impacts on climate and control the distribution of nutrients, heat uptake, and carbon cycling (Hogg et al., 2015). Eddies also play an important role in the comprehensive response of the climate system to warming (Hewitt et al. 2022; Rackow et al. 2022, Griffies et al. 2015). In addition to the influence of mesoscale ocean features on the predictability of European weather downstream of the Gulf Stream area (Keeley et al., 2012), it has been proposed that higher-resolution simulations can enhance the representation of local heterogeneities in the sea-ice cover (Hutter et al., 2022). Via their impact on small-scale ocean features such as eddies, atmospheric storms can impact deep water formation in the Labrador Sea (Gutjahr et al. 2022), an ocean region of global significance because of its role in the meridional overturning circulation of the ocean. Coupled ocean-atmosphere variability patterns such as the El Nino-Southern Oscillation (ENSO), the largest signal of interannual variability on Earth, may also benefit from km-scale resolutions since ENSO-relevant ocean meso-scale features (Wengel et al. 2021) as well as westerly wind bursts should be better resolved.

High-resolution simulations pose significant challenges in terms of numerical methods, data management, storage and analysis (Schär et al., 2020). To exploit the potential of km-scale modelling, it is essential to develop scalable models that can run efficiently on large supercomputers and take advantage of the next generation of exascale computing platforms (Bauer et al.,



2021, Taylor et al., 2023). Global atmosphere-only climate simulations at km-scale were pioneered by the NICAM group
(Nonhydrostatic ICosahedral Atmospheric Model) almost two decades ago. On sub-seasonal to seasonal time scales, a global
aqua-planet configuration at 3.5km resolution was performed (Tomita et al. 2005), and the MJO was realistically reproduced
at 7km and 3.5km resolutions (Miura et al. 2007). In the last decade, the NICAM group as well as the European Centre for
Medium-Range Weather Forecasts (ECMWF) ran simulations on climate time scales around 10-15km resolution. In particular,
14km resolution 30-year AMIP (Kodama et al. 2015) and HighResMIP simulations (Kodama et al. 2021) were performed with
NICAM. During Project Athena, the sensitivity to horizontal resolution of ECMWF's Integrated Forecasting System in terms
of climate and seasonal predictive skill has been analysed at resolutions up to 10km via many 13 months simulations (totalling
several decadal simulations), and with help of a 48-year AMIP-style simulation plus future time slices at 15km resolution
(Jung et al. 2012). Recently, the NICAM group presented 10-year AMIP simulations at 3.5km using an updated NICAM
version (Takasuka et al. 2024). Other modelling groups around the world have also increased their model resolution towards
the km-scale, and many participated in the recent DYAMOND intercomparison project (DYnamics of the Atmospheric general
circulation Modeled On Non-hydrostatic Domains) with a grid spacing as fine as 2.5km, simulations running over 40 days,
and some of them already coupled to an ocean (Stevens et al., 2019).

While different modelling groups push global atmosphere-only simulations towards unprecedented resolutions, e.g. 220m
resolution in short simulations with NICAM, another scientific frontier has emerged around running km-scale simulations on
multi-year timescales, coupled to an equally refined ocean model. Indeed, in the last years, there have been several examples
of km-scale simulations that have been run on up to monthly and seasonal timescales (Stevens et al., 2019, Wedi et al., 2020),
but not many beyond these timescales, and not yet with a km-scale ocean (Miyakawa et al. 2017). This is due to the fact that
even the most efficient high-resolution coupled models are computationally expensive and require substantial resources to run.
This in turn limits the number of simulations and realisations that can be performed, making it difficult to calibrate and optimise
the model settings. Coarser resolution models have been tuned for decades to be relatively reliable on the spatial scales that
they can resolve, and to match the historical period well for which high-quality observations are available. Nevertheless, this
is often achieved by compensating errors, which can not necessarily be expected to compensate in a warming climate. These
models also have some long-standing biases that can locally be larger than the interannual variability or the climate change
signal (Rackow et al. 2019, Palmer and Stevens, 2019). The lack of explicitly simulated small-scale features is one potential
source for these long-standing biases in weather and climate models (Schär et al., 2020). Coarser resolution models also
struggle with answering some important climate questions, such as what will happen to extreme events in a warmer world, and
with providing accurate information on climate changes at regional scale.

The European H2020 Next Generation Earth Modelling Systems (nextGEMS) project aims to build a new generation of eddy-
and storm-resolving global coupled Earth System Models to be used for multi-decadal climate projections at km-scale. By
providing globally consistent information at scales where extreme events and the effects of climate change matter and are felt,





global km-scale multi-decadal projections will support the increasing need to provide localised climate information to inform local adaptation measures. The nextGEMS models build upon models that are also operationally used for numerical weather prediction (NWP): ICON, which is jointly developed by DWD and MPI-M (Hohenegger et al., 2023), and the Integrated Forecasting System (IFS) of ECMWF, coupled to the NEMO and FESOM ocean models. NextGEMS revolves around a series of hackathons, in which the simulations performed with the two models are examined in detail by an international community of more than 100 participants, followed by new model development iterations or 'Cycles'. The nextGEMS models have been (re-)designed for scalability and portability across different architectures (Satoh et al. 2019, Schulthess et al. 2019, Müller et al. 2019, Bauer et al. 2020, Bauer, Quintino, and Wedi 2022) and lay the foundation for the Climate Change Adaptation Digital Twin developed in the EU's Destination Earth initiative (DestinE).

The operational NWP system at ECMWF uses an average 9km grid-spacing for the atmosphere coupled to an ocean at 0.25° spatial resolution (NEMO v3.4), which translates to a horizontal grid spacing of about 25km along the equator. While many coupled effects such as the atmosphere-ocean interactions during tropical cyclone conditions (Mogensen et al. 2017) can be realistically simulated at this resolution, ocean eddies in the mid latitudes are still only 'permitted' due to their decreasing size with latitude (Hallberg 2013). This setup is far from our goal to explicitly resolve mesoscale ocean eddies all around the globe (Sein et al., 2017). In this study, we therefore focus mainly on two configurations in which km-scale versions of IFS (one at 2.8km and one at 4.4km grid spacing in the atmosphere and land) are coupled to the FESOM2.5 ocean-sea ice model at about 5km grid spacing, developed by the Alfred Wegener Institute, Helmholtz Centre for Polar and Marine Research (AWI). These configurations allow us to resolve many essential climate processes directly, for example mesoscale ocean eddies and sea ice leads in large parts of the mid- and high-latitude ocean, atmospheric storms, as well as certain small-scale features in the topography and land surface. We also test new developments of the IFS carried out in the last years at ECMWF to improve the representation of snow cover, land surface, and cities world-wide.

This paper documents the coupled km-scale model configurations with the Integrated Forecasting System in Section 2. The technical and scientific model improvements, carried out along the nextGEMS model development cycles based on feedback by the nextGEMS community, are presented in Section 3. A first set of emerging advances stemming from the km-scale character of the simulations is presented in Section 4, and more in-depth process studies will be the focus of dedicated future work. The paper closes with a summary and discussion of future steps in Section 5.

## 2 Model configurations

### 2.1 The Integrated Forecasting System and its coupling to NEMO and FESOM

The Integrated Forecasting System (IFS) is a spectral-transform atmospheric model with two-time-level semi-implicit, semi-Lagrangian time-stepping (Temperton et al., 2001; Hortal, 2002; Diamantakis and Váňa, 2022). It is coupled to other Earth





System components (land, waves, ocean, sea-ice), and it is used in its version 48r1 (https://www.ecmwf.int/en/publications/ifs-
documentation, last access 26 March 2024), which has been used for operational forecasts at ECMWF since July 2023 (plus
modifications that will be detailed in this study). In its operational configuration ('oper'), the atmospheric component is
coupled to the NEMO v3.4 ocean model. A Gaussian octahedral grid (TCo) is used for the atmospheric grid-point calculations
(advection, physical parametrizations, product of terms), which implies higher effective resolution and better efficiency than
other standard (linear) Gaussian grids. It acts as a numerical filter without the need for expensive de-aliasing procedures,
requires little diffusion, and produces small total mass conservation errors for medium-range forecasts (Wedi 2014, Malardel
et al., 2016). A hybrid, pressure-based vertical coordinate is used which is a monotonic function of pressure and depends on
the surface pressure (Simmons and Strüfing, 1983). The vertical coordinate follows the terrain at the lowest level and relaxes
to a pure pressure-level vertical coordinate system in the upper part of the atmosphere. The vertical discretization scheme is a
finite element method using cubic B-spline basis functions (Vivoda et al., 2018, Untch and Hortal, 2004).

The atmosphere component of the IFS has a full range of parametrizations described in detail in ECMWF (2023a,b). The moist
convection parameterization, originally described in Tiedtke (1989), is based on the mass-flux approach, and represents deep,
shallow and mid-level convection. For deep convection the mass-flux is determined by removing a modified Convective
Available Potential Energy (CAPE) over a given time scale (Bechtold et al., 2008, 2014), taking into account an additional
dependence on total moisture convergence and a grid resolution dependent scaling factor to reduce the cloud base mass flux
further at grid resolutions higher than 9km (Becker et al., 2021). The sub-grid cloud and precipitation microphysics scheme is
based on Tiedtke (1993) and has since been substantially upgraded with separate prognostic variables for cloud water, cloud
ice, rain, snow and cloud fraction, and an improved parametrization of microphysical processes (Forbes et al. 2011; Forbes
and Ahlgrimm, 2014). The parametrization of sub-grid turbulent mixing follows the Eddy-Diffusivity Mass-Flux (EDMF)
framework, with a K-diffusion turbulence closure and a mass-flux component to represent the non-local eddy fluxes in unstable
boundary layers (Siebesma et al., 2007; Kohler et al., 2011). The orographic gravity wave drag is parametrized following Lott
and Miller (1997) and Beljaars et al. (2004) and a non-orographic gravity wave drag parametrization is described in Orr et al.
(2010). The radiation scheme is described in Hogan and Bozzo (2018, ecRad). Full radiation computations are calculated on a
reduced resolution grid every hour with approximate updates for radiation-surface interactions every timestep at full resolution.

The land model of the IFS is ECLand (Bousetta et al. 2021), which runs on the identical grid and is fully coupled to the
atmosphere through an implicit flux solver. ECLand represents the surface processes that interact with the atmosphere in the
form of fluxes. The ECLand version used in this work currently contains among others, a 4-layer soil scheme, a lake model,
an urban model, a simple vegetation model, a multi-layer snow scheme, and a vast range of global maps describing the surface
characteristics. A wave model component is provided by ecWAM to account for sea state dependent processes in the IFS
(ECMWF, 2023c). The wave model runs on a reduced lat-lon grid with 0.125° resolution, 36 frequencies, and 36 directions.
This means that the distance between latitudes is 0.125°, and the number of points per latitude is reduced polewards in order



to keep the actual distance between grid points roughly equal to the spacing between two consecutive latitudes. The frequency
discretisation is such that ocean waves with periods between 1 and 28 seconds are represented.

For the purpose of nextGEMS and other related projects such as the DestinE Climate Change Adaptation Digital Twin, where
also an IFS-NEMO configuration with a 1/12 degree ocean (NEMO v4) is applied, the complementary IFS-FESOM model
option was developed. We coupled the Finite VolumE Sea ice-Ocean Model FESOM2 (Danilov et al. 2017, Scholz et al. 2019,
Koldunov et al. 2019, Sidorenko et al. 2019) to IFS (see details below). Instead of using a coupler for this task, as for the
OpenIFS-FESOM (Streffing et al. 2022), the alternative adopted here is to follow the strategy for IFS-NEMO coupling, where
the ocean and IFS models are integrated into a single executable and share a common time stepping loop (Mogensen, Keeley,
and Towers, 2012). In this sequential coupling approach (akin to the model physics-dynamics and land-surface coupling that
occurs every model timestep), the atmosphere advances for 1 hour (length of the coupling interval) and fluxes are passed as
upper boundary condition to the ocean, which then in turn advances for 1 hour, up to the same checkpoint. The following
atmospheric step then uses updated surface ocean fields as lower boundary condition for the next coupling interval (Mogensen,
Keeley, and Towers, 2012). Note that there is no need to introduce a lag of one coupling timestep because the ocean and
atmosphere models run sequentially and not overlapping in parallel. A study into the effect of model lag on flux/state
convergence by Marti et al. (2021) found that sequential instead of parallel coupling reduces the error nearly to the fully
converged solution.

In the operational IFS, in areas where sea ice is present in the ocean model, currently a sea ice thickness of 1.5m and no snow
cover are assumed for the computation of the conductive heat flux on the atmospheric side. Our initial implementation for the
multi-year simulations carried out in nextGEMS does not divert yet from this assumption of the operational configuration, in
which the atmosphere 'sees' only the sea-ice fraction computed by the ocean/sea-ice model. There are more consistent options
available to couple the simulated sea ice albedo, ice surface temperature, ice and snow thickness from the ocean models to the
atmospheric component (Mogensen, Keeley, and Towers, 2012) and those will also be considered in future setups.

In order to run the IFS on different High Performance Computing (HPC) architectures, we adopted ECMWF's "RAPS" (Real
Applications on Parallel Systems) benchmark configuration. The RAPS code has been further adopted for use as a running
environment for multi-annual runs. RAPS is a set of mostly shell scripts that allow to build, configure, and submit coupled IFS
experiments on external HPC hardware, other than ECMWF's on-premise Atos computer in Bologna, Italy. In this
environment, FESOM can be linked as an external ocean library to the IFS. This allows the IFS to control the ocean as a
subroutine and to trigger the ocean initialisation, time-stepping, and finalisation steps (see Section 2.2).



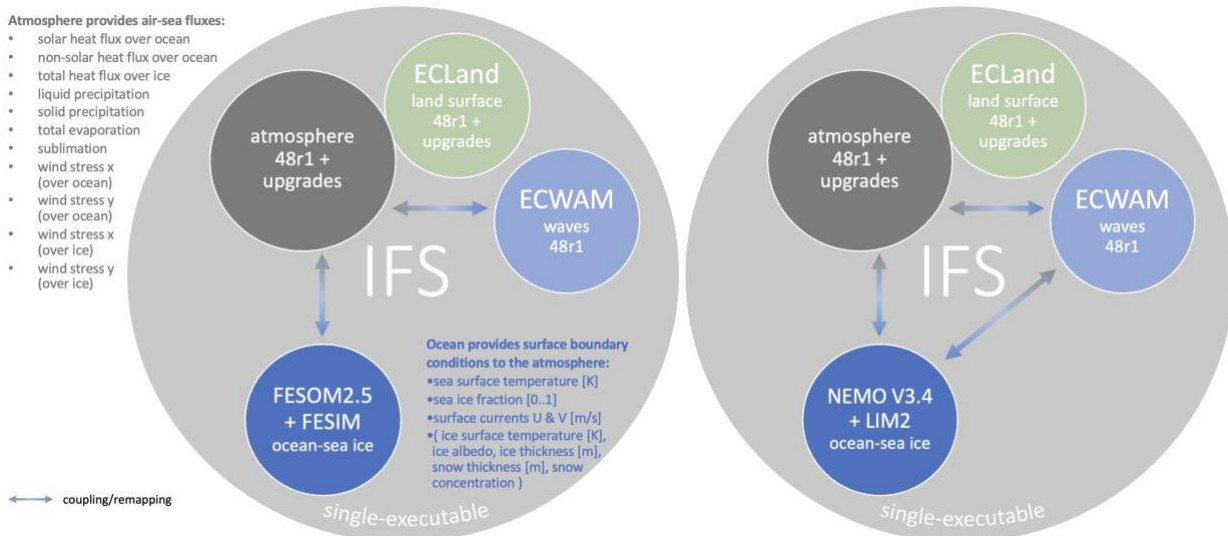


**Figure 1: Coupling of the Integrated Forecasting System (IFS) components in (left) IFS-FESOM and (right) IFS-NEMO in nextGEMS configurations.** Coupling between the unstructured FESOM grid and the Gaussian grid of the atmosphere is via pre-prepared remapping weights in SCRIP format (Jones 1999). Direct coupling between the surface wave model (ECWAM) and the ocean is at the moment only implemented in IFS-NEMO; in IFS-FESOM, the ocean and waves interact only indirectly via the atmosphere. ECWAM and the atmosphere have their own set of remapping weights for direct coupling, while ECLand and the atmosphere are more closely coupled to each other.

The oceans provide surface boundary conditions to the atmosphere (sea surface temperature, sea ice concentration, zonal and meridional surface currents) while the atmospheric component provides air-sea fluxes to the ocean models (as listed in Fig. 1). The exchange between the different model grids is implemented as a Gaussian distance-weighted interpolation for both directions. Since the implementation accepts any weight files as long as they are provided in SCRIP format (Jones 1999), future setups will explore other interpolation strategies, such as the use of conservative remapping weights for the air-sea fluxes to ensure better flux conservation. River runoff for the ocean models is taken from climatology; for IFS-FESOM, the runoff from the COREv2 (Large and Yeager 2009) flux dataset is applied based on Dai et al. (2009).

In order to couple FESOM with IFS, the existing single-executable coupling interface (i.e. the set of Fortran subroutines) between IFS and NEMO (Mogensen, Keeley, and Towers, 2012) has been extracted and newly implemented directly in the FESOM source code (Rackow et al. 2023c). From the perspective of the atmospheric component, after linking, FESOM and NEMO thus appear to IFS virtually identical in terms of provided fields and functionality in forecast runs with IFS. Clear gaps and differences to the operational configuration with NEMO v3.4 remain in terms of ocean data assimilation capabilities (NEMOVAR), ocean initial condition generation, and missing surface ocean-wave coupling (Fig. 1). However, these differences do not critically impact the multi-year simulations for nextGEMS described in this study or multi-decadal simulations planned for nextGEMS and Destination Earth.



**2.2 Technical refactoring for the FESOM2.5 ocean-sea ice model code**

Prior to the start of nextGEMS, FESOM had been fully MPI-parallelised only and was shown to scale well on processor counts beyond 100,000 (Koldunov et al. 2019). In the nextGEMS configuration with IFS, which is set up in a single-executable environment with a hybrid MPI-OpenMP parallelisation approach, experience from Cycle 1 had shown that it had become necessary to rewrite the ocean model code to fully support hybrid MPI-OpenMP parallelization for the planned multi-year high-resolution simulations in order for the ocean to make use of the available resources to full extent. In detail, this meant rewriting numerous non-iterative loops in the ocean model code to support OpenMP parallelisation with release of FESOM version 2.5.

The FESOM model has been significantly refactored also in other aspects over the last years to support coupling with IFS. In the single executable coupled system, the IFS initializes the MPI communicator (Mogensen, Keeley, and Towers, 2012) and passes it to the ocean model for initialisation of FESOM. In particular, FESOM's main routine has been split into 3 cleanly defined steps, namely the initialisation, time stepping, and finalisation steps. This was a necessary step for the current single-executable coupled model strategy at ECMWF, where the ocean is called and controlled from within the atmospheric model. The single-executable configuration is a necessary condition for coupled data assimilation at ECMWF. The adopted strategy means that some IFS-NEMO developments can be directly applied also to IFS-FESOM configurations. Similar to what is done for the wave and atmosphere components of the IFS, we implemented a fast "memory dump" restart mechanism for FESOM. This has the advantage that the whole coupled model can be quickly restarted as long as the parallel distribution (number of MPI tasks and OpenMP processes) does not change during the simulation.

**2.3 Model output and online diagnostics**

One of the concerns for the scientific evaluation of multi-year high-resolution simulations is the need to read large volumes of output from the global parallel filesystem. This is required for certain processing tasks, such as the computation of monthly averages in a climate context and regridding to regular meshes, so that the relevant information can be easily analysed and visualised. One way to mitigate this burden is to move these computations closer to where the data is produced and process the data in memory. Many of these computations are currently not possible in the IFS code, so we used MultIO (Sármány et al., 2023), a set of software libraries that provide, among other functionalities, user-programmable processing pipelines that operate on model output directly. IFS has its own Fortran-based I/O-server that is responsible for aggregating geographically distributed three-dimensional information and creating layers of horizontal two-dimensional fields. It passes these pre-aggregated fields directly to MultIO for the on-the-fly computation of temporal means and data regridding.

One of the key benefits of this approach is that with the in-memory computation of, for example, monthly statistics, the



requirement of storage space may be reduced significantly. Higher-frequency data may only be required for the computation
of these statistics and as such would not need to be written to disk at all. For the nextGEMS runs in this study, however, the
decision was taken to make use of MultIO for user-convenience mostly, i.e. to produce post-processed output in addition to
the native high-frequency output. The computational overhead associated with this (approximately 15% in this case) is more
than offset by the increased productivity gained from much faster and easier evaluation of high-resolution climate output,
particularly in the context of hackathons with a large number of participants. As a result, the MultIO pipelines have been
configured to support the following five groups of output:
● Hourly or six-hourly output (depending on variable) on native grids.
● Hourly or six-hourly output (depending on variable), interpolated to regular (coarser) meshes for ease of data analysis.
The MultIO configuration uses parts of the functionality of the Meteorological Interpolation and Regridding package
(MIR), ECMWF's open-source re-gridding software, to be able to execute this in memory.
● Monthly means for all output variables on native grids.
● Monthly means for all output variables on regular (coarser) meshes, interpolated by MultIO calling MIR.
● All fields are encoded or re-encoded in GRIB by MultIO calling ECCODES, an open-source encoding library.
At the end of each pipeline, all data are streamed to disk, more specifically to the Fields DataBase (FDB, Smart et al., 2017),
an indexed domain-specific object store for archival and retrieval – according to a well-defined schema – of meteorological
and climate data. This mirrors the operational setup at ECMWF. For the nextGEMS hackathons, all simulations and their
GRIB data in the corresponding FDBs have been made available in Jupyter notebooks (Kluyver et al. 2016) via intake catalogs
(https://intake.readthedocs.io/en/latest/, last access 25 March 2024) using gribscan. The gribscan tools scans GRIB files and
creates Zarr-compatible indices (Kölling, Kluft, and Rackow, 2024).

## 2.4 Performed nextGEMS runs

The nextGEMS project relies on several model development cycles, in which the high-res models are run and improved based
on community feedback from the analysis of successive runs. In an initial set of km-scale coupled simulations (termed 'Cycle
1'), the models were integrated for 75 days, starting on 20 January 2020 (Table 1). For Cycle 1, ECMWF's IFS has been run
at 9km (TCo1279 in Gaussian octahedral grid notation) and 4.4km (TCo2559) global spatial resolution. The runs at 9km were
performed with the deep convection parametrization, while at 4.4km, the IFS was run with and without the deep convection
parametrization. The underlying ocean models NEMO and FESOM2.1 had been run on an eddy-permitting 0.25° resolution
grid in this initial model cycle (ORCA025 for NEMO and a triangulated version of this for FESOM, tORCA025). Based on
the analysis by project partners during a hackathon organised in Berlin in October 2021, several key issues were identified
both in the runs with IFS, and in those run with ICON (Hohenegger et al. 2023).



As will be detailed below, the IFS has been significantly improved for the longer 'Cycle 2' simulations (IFS nextGEMS Cycle 2 4.4km 1-year simulation, https://dx.doi.org/10.21957/1n36-qg55; Wieners et al., 2023), where a 2.8km simulation (TCo3999) has also been performed. For the purpose of nextGEMS Cycle 2 and 3, an ocean grid with up to 5km resolution ('NG5') has been introduced for the FESOM model, which is eddy-resolving in most parts of the global ocean (see Appendix B). The NG5 ocean has been spun up for a duration of 5 years in stand-alone mode, with ERA5 atmospheric forcing (Hersbach et al. 2020) until 20 January 2020. In contrast, NEMO performs active data assimilation to estimate ocean initial conditions for 20 January 2020.

Based on feedback from the 2nd hackathon in Vienna in 2022, 'Cycle 3' simulations for the recent 3rd hackathon in Madrid (June 2023) have been further improved. The ocean has been updated to FESOM2.5 (Rackow et al, 2023c), and run coupled for up to 5 years (see Fig. 2 for an example wind speed snapshot at 4.4km resolution). In the following section, we will detail the series of scientific improvements in the atmosphere, ocean, and land components of IFS-NEMO/FESOM that were performed to address the identified key issues, and how these successive steps result in a better representation of the coupled physical system.

**Table 1: nextGEMS configurations of the IFS and coupled simulations analysed in this study.** The Gaussian octahedral grid notations TCo1279, TCo2559, and TCo3999 refer to 9km, 4.4km, and 2.8km global atmospheric spatial resolution, respectively. The simulations were performed with constant greenhouse gas forcing from the year 2020 ($CO_2$ = 413.72 ppmv, $CH_4$ = 1914.28 ppbv, $N_2O$ = 331.80 ppbv, CFC11 = 857.38 pptv, CFC12 = 497.10 pptv), prognostic ozone, no volcanic aerosols, and the CAMS aerosol climatology (Bozzo et al. 2020).

| Configuration | Length of simulations | Ocean settings |
|---|---|---|
| IFS-NEMO, TCo1279 ('oper')<br><br>Cycle 3, 2, 1 | 5 years (Cycle 3)<br>2 years (Cycle 2)<br>75 days (Cycle 1) | NEMO V3.4, ORCA025<br>(0.25° 3-polar grid) |
| IFS-FESOM, TCo1279-NG5<br><br>Cycle 3 | 1 year (Cycle 3) | FESOM2.5, NG5 grid (3-4km in high-res regions; 13km in tropics) |
| IFS-FESOM, TCo3999-NG5<br><br>Cycle 2 | 8 months (Cycle 2) | FESOM2.1, NG5 grid (3-4km in high-res regions; 13km in tropics) |
| IFS-FESOM, TCo2559-NG5<br><br>Cycle 3 & 2 | 5 years (Cycle 3)<br>1 year (Cycle 2) | FESOM2.1/2.5, NG5 grid (3-4km in high-res regions; 13km in tropics) |



305

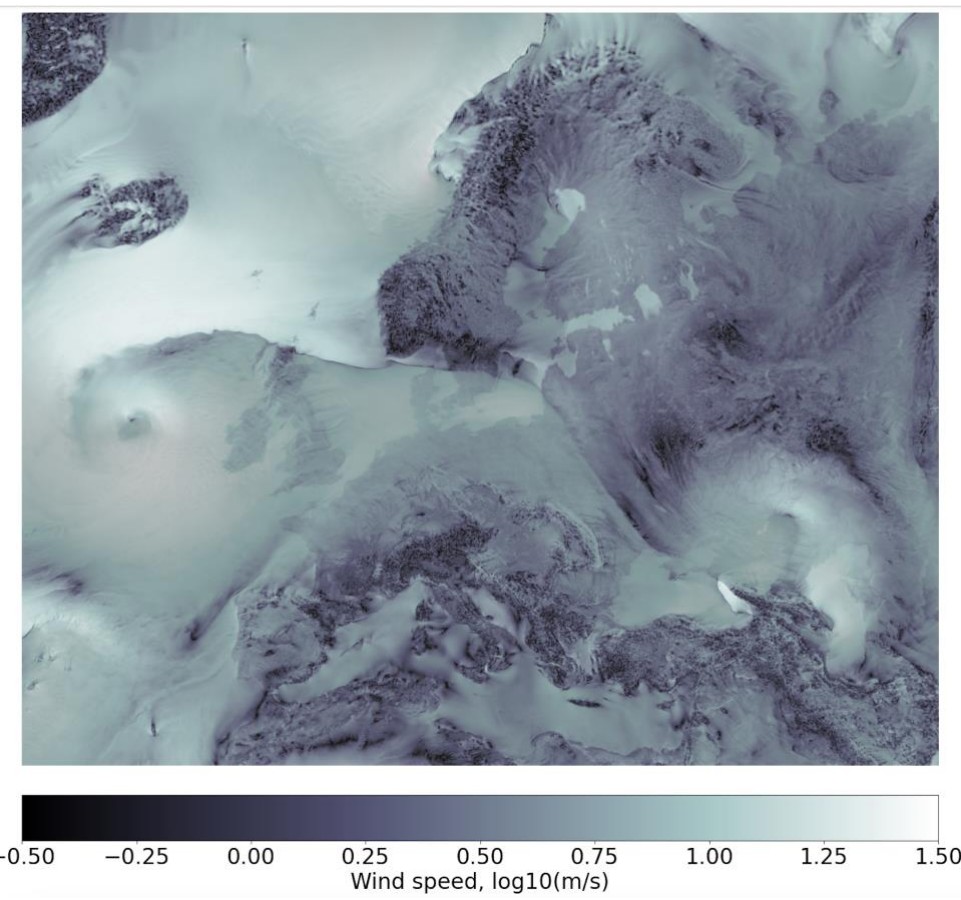

306

**Figure 2: Wind speed snapshot over Europe as simulated by IFS with a 4.4km spatial resolution in the atmosphere.**
The wind speed map is overlaid with a map of the zonal wind component in a grey-scale colormap for further shading, which
is made partly transparent. The figure does not explicitly plot land. The high-resolution simulation nevertheless clearly shows
the continental land masses and orographic details due to larger surface friction and hence smaller wind speeds (darker areas
depict lower wind speeds). The image is a reproduction with Cycle 3 data of the award winning entry by N. Koldunov for
2022's Helmholtz Scientific Imaging Contest https://helmholtz-imaging.de/about_us/overview/index_eng.html).

## 3 Model developments for multi-year coupled km-scale IFS simulations

This section details model developments for the atmosphere (3.1), ocean and sea ice (3.2), wave (3.3), and land (3.4)
components of IFS-NEMO/FESOM in the different cycles of nextGEMS. Following a short overview of identified key issues
and developments at the beginning of each section, we present how those successive development steps translate to a better
representation of the coupled physical system.



### 3.1 Atmosphere

### 3.1.1 Key issues and model developments

**Water and energy imbalances**

At the first nextGEMS hackathon, strong water and energy imbalances were identified as key issues in the Cycle 1 simulations, which led to large biases in the top-of-atmosphere (TOA) radiation balance. If run for longer than 75 days, e.g. multiple years, this would lead to a strong drift in global mean 2m temperature. Analysis confirmed that most of the energy imbalance in the IFS was related to water non-conservation, and that this issue gets worse (i) when spatial resolution is increased, and (ii) when the parametrization of deep convection is switched off (hereafter 'Deep Off'). This is because the semi-Lagrangian advection scheme used in the IFS is not conserving the mass of advected tracers, e.g. the water species (see Appendix A). However, while this issue was acknowledged to be detrimental for the accuracy of climate integrations, so far it was thought that it was small enough to not significantly affect the quality of numerical weather forecasts, which span timescales ranging from a few hours to seasons ahead. To address the problem of water non-conservation in the IFS, a tracer global mass fixer was activated for all prognostic hydrometeors (cloud liquid, ice, rain and snow) in nextGEMS Cycle 2, as well as water vapour (for more details, see Appendix A describing the mass-fixer approach). The tracer mass fixer ensures global mass conservation, but it cannot guarantee local mass conservation. However, it estimates where the mass conservation errors are larger and inserts larger corrections in such regions, which is often beneficial for local mass conservation and accuracy (see Diamantakis and Agusti-Panareda, 2017). When adding tracer mass fixers to a simulation, the computational cost increases by a few percentage points (typically less than 5%). Water and energy conservation in Cycle 1 versus Cycle 2 is discussed in Section 3.1.2.

**Top-of-atmosphere radiation balance**

To reduce drift in global mean surface temperature, it is essential that the global top-of-atmosphere (TOA) radiation imbalance is small. In the nextGEMS Cycle 2 simulation at 4.4 km resolution coupled to FESOM2.1 (Table 1), the TOA net imbalance, relative to observed fluxes from the CERES-EBAF product (Loeb et al. 2018), had been about +3 Wm$^{-2}$ (positive values indicate downward fluxes), resulting from a +5 Wm$^{-2}$ shortwave imbalance that was partly balanced by a -2 Wm$^{-2}$ longwave imbalance. Because of anthropogenic greenhouse gas emissions, CERES shows a +1 Wm$^{-2}$ imbalance. Due to the larger TOA imbalance, the nextGEMS Cycle 2 simulations warmed too much, by about 1K over the course of one year (see Section 3.1.3). Thus, addressing the TOA radiation imbalance was a major development focus in preparation for the 5-year integration in nextGEMS Cycle 3.

On top of IFS 48r1, in Cycle 3 we used a combination of model changes targeting a reduced TOA radiation imbalance, mostly affecting cloud amount. Changes that increased the fraction of low clouds are (i) a change restricting the detrainment of mid-level convection to the liquid phase, (ii) a reduction of cloud edge erosion following Fielding et al. (2020) and (iii) a reduction of the cloud inhomogeneity, which increases cloud amount as it reduces the rate of accretion. This change is in line with nextGEMS's km-scale resolutions as cloud inhomogeneity is expected to be smaller at high resolutions. High clouds were



increased in areas with strong deep convective activity by (iv) decreasing a threshold that limits the minimum size of ice
effective radius, in agreement with observational evidence and (v) changing from cubic to linear interpolation for the departure
point interpolation of the Semi-Lagrangian advection scheme for all moist species except water vapour. The resulting TOA
balance in Cycle 3 is discussed in Section 3.1.3.

**Representation of intense precipitation and convective cells**

Precipitation has many important roles in the climate system. It is not only important for the water cycle over land and ocean,
but also provides a source of energy to the atmosphere, as heat is released when water vapour condensates and rain forms,
which balances radiative cooling. Precipitation is also often associated with meso-scale or large-scale vertical motion and the
corresponding overturning circulation is crucial for the horizontal and vertical redistribution of moisture and energy within the
atmosphere.
In km-scale simulations in which the deep convection parametrization is switched off (e.g, Cycle 2 at 4.4 km and 2.8 km
resolution), convective cells tend to be too localised, too intense, and they lack organisation into larger convective systems
(e.g, Crook et al., 2017, Becker et al., 2021). The characteristics of meso-scale organisation of convection also affect the larger
scales, for instance zonal mean precipitation and the associated large-scale circulation. For example, with deep convection
parametrization off in Cycle 2 (Deep Off), the ITCZ often organises into a continuous and persistent line of deep convection
over the Pacific at 5°N (see Fig. D1 in Appendix D), and the zonal mean precipitation at 5°N is strongly overestimated.
To address these issues, instead of switching the deep convection scheme off completely, we have reduced its activity by
reducing the cloud-base mass flux in Cycle 3. The cloud-base mass flux is the key ingredient of the convective closure, and
depends on the convective adjustment time scale $\tau$, which assures a transition to resolved convection at high resolution via an
empirical scaling function that depends on the grid spacing (discussed in more detail in Becker et al., 2021). To significantly
reduce the activity of the deep convection scheme in Cycle 3, we use the value of the empirical scaling function that is by
default used at 700m resolution (TCo15999) already at 4.4 km resolution (TCo2559), which corresponds to a reduction of the
empirical value that determines the cloud base mass flux by a factor of 6 compared to its value at 9 km resolution. Precipitation
characteristics in Cycle 3 vs Cycle 2 are discussed in Section 3.1.4.

**3.1.2 Improvements of mass and energy conservation in Cycle 2 vs Cycle 1**

To address the water non-conservation mentioned in Section 3.1.1, tracer mass fixers for all moist species were introduced in
Cycle 2. Figure 3 shows that the Cycle 1 simulations with the IFS have an artificial source of water in the atmosphere. This
artificial source is responsible for 4.6% of total precipitation in the 9 km simulation with deep convection parametrization
switched on (hereafter 'Deep On'), which is also used for ECMWF's operational high-resolution ten-day forecasts, and for
10.7% at 4.4 km with Deep Off. Further analysis after the hackathon by the modelling teams at ECMWF has shown that about
50% of the artificial atmospheric water source is created as water vapour. The additional water vapour not only affects the



radiation energy budget of the atmosphere, but it can also cause energy non-conservation when heat is released through
condensation. The other 50% of water is created as cloud liquid, cloud ice, rain or snow. This is related to the higher-order
interpolation in the semi-Lagrangian advection scheme introduced for cloud liquid, cloud ice, rain and snow in IFS Cycle 47r3,
which can result in spurious maxima and minima, including negative values, which are then clipped to remain physical. It
turns out that the spurious minima are in excess of the spurious maxima and by clipping them, the mass of cloud liquid, cloud
ice, rain and snow is effectively increased. When activating global tracer mass fixers, global water non-conservation is
essentially eliminated (about 0.1%) in the Cycle 2 simulations (Figure 3).

On a global scale, the total energy budget of the atmosphere can be defined as
$$\frac{c_{\mathrm{pd}}}{g} \int_{p_{\mathrm{surf}}}^{0} \frac{dT}{dt} dp_{\mathrm{h}} + \frac{L_{v0}}{g} \int_{p_{\mathrm{surf}}}^{0} \frac{dq_v}{dt} dp_{\mathrm{h}} - \frac{L_{s0} - L_{v0}}{g} \int_{p_{\mathrm{surf}}}^{0} \frac{dq_{\mathrm{i}} + dq_{\mathrm{s}}}{dt} dp_{\mathrm{h}} + \int_{p_{\mathrm{surf}}}^{0} \frac{dKE}{dt} dp_{\mathrm{h}}$$

$$= F_{\mathrm{s}} + F_{\mathrm{q}} - F_{\mathrm{rad}}^{\mathrm{top}} + F_{\mathrm{rad}}^{\mathrm{surf}} + (L_{s0} - L_{v0})P_{\mathrm{s}} ,$$


where $T$ is temperature, $q_{\mathrm{v}}$, $q_{\mathrm{l}}$ and $q_{\mathrm{i}}$ are water vapour, cloud ice and snow. Together, these terms describe the change in
vertically-integrated frozen moist static energy over time, while the last term on the left-hand-side of the equation is the change
in vertically-integrated kinetic energy ($KE$). Sources and sinks of the atmosphere's total energy are $F_{\mathrm{s}}$ and $F_{\mathrm{q}}$, which are the
surface turbulent sensible and latent heat fluxes, $F_{\mathrm{rad}}^{\mathrm{top}}$ and $F_{\mathrm{rad}}^{\mathrm{surf}}$, which are the TOA and surface net radiative shortwave and
longwave fluxes, and $(L_{s0}-L_{v0})P_{\mathrm{s}}$ is the energy required to melt snow at the surface. Note that dissipation is not a source or sink
of total energy.

Using this equation to calculate the global energy budget imbalance in Figure 3, the Cycle 1 simulation with 9 km resolution
has an atmospheric energy imbalance of 2.0 Wm-$^{-2}$, and this imbalance increased to 6.4 Wm$^{-2}$ at 4.4 km resolution with Deep
Off. In Cycle 2, the energy budget imbalance due to the mass conservation of water species is substantially smaller, having
reduced to less than 1 Wm$^{-2}$. This remaining imbalance can be related to the explicit and semi-implicit dynamics because they
are still non-conserving, for example causing an error in surface pressure, as well as the mass fixers. The remaining imbalance
could be removed by adding a total energy fixer to the model. The discussed setup with improved water and energy
conservation is part of ECMWF's recent operational IFS upgrade in June 2023 (48r1) because it improves the skill scores of
the operational weather forecasts (ECMWF Newsletter 172, 2022).





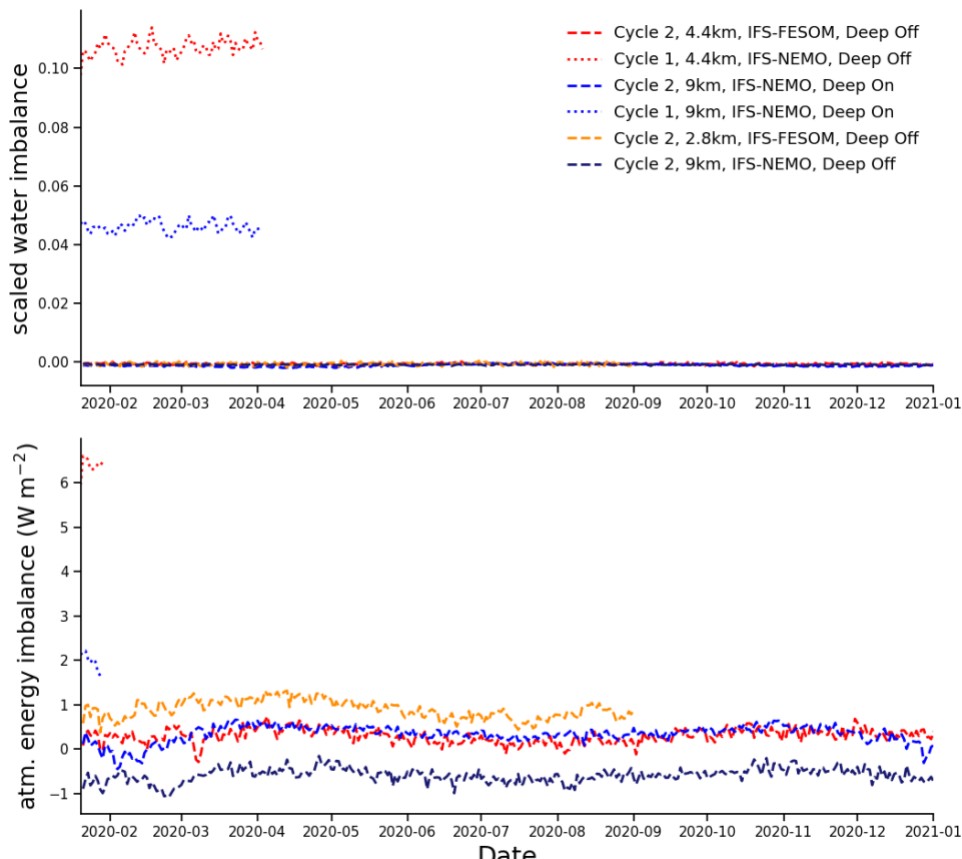

**Figure 3: Daily mean water non-conservation (left) and daily-mean atmospheric energy imbalance (right), as a function of lead time for Cycle 1 and Cycle 2 simulations.** Water non-conservation is computed as the daily change in globally integrated total water, taking account of surface evaporation and precipitation, as a fraction of the daily precipitation. The atmospheric energy imbalance is calculated with Equation 1.

### 3.1.3 Realistic TOA radiation balance and surface temperature evolution in Cycle 3

Due to the model changes detailed in Section 3.1.1, the nextGEMS Cycle 3 simulations with the IFS have at all resolutions a TOA radiation imbalance that is within observational uncertainty, with respect to the net, shortwave and longwave fluxes (Figure 4). This is not only true for the annual mean value, but also for the annual cycle of TOA imbalance (8-shape in Figure 4).

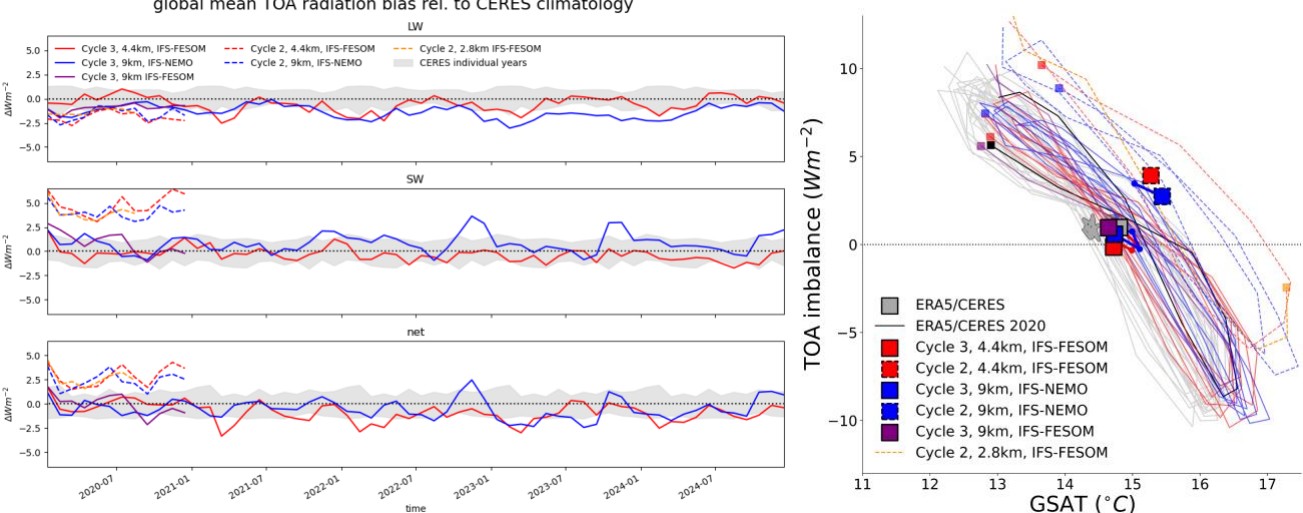

**Figure 4: Global-mean TOA radiation deviation from the CERES climatology in the 5-year-long nextGEMS simulations and global-mean TOA imbalance as function of global-mean surface air temperature (GSAT).** a) Grey shading shows the climatological range of individual CERES years. Due to the free-running nature of the nextGEMS simulations, variations within the grey envelope are to be expected even in the absence of any bias. b) Grey lines show the climatological range of individual CERES years (2001–2020) over ERA5 GSAT data (Hersbach et al. 2020). Thin lines are tracing monthly mean values with a small square marking the final month for each simulation. Big squares depict annual means (dashed for Cycle 2, solid for Cycle 3) and for multi-year simulations thick solid lines are tracing annual means for each year with the big square marking the last simulated annual mean.

As a result, the global mean surface temperature in the Cycle 3 simulations is in close agreement with the ERA5 reanalysis (Hersbach et al. 2020), and stays in close agreement over the 5 years of coupled simulations (Figure 5 and Figure C1 in Appendix C).

Going from Cycle 2 to Cycle 3, the warming over time is not evident anymore in IFS-FESOM and IFS-NEMO (Figure 5). Differences in local warming over the Southern Ocean in the two models are further discussed in section 3.2.2.





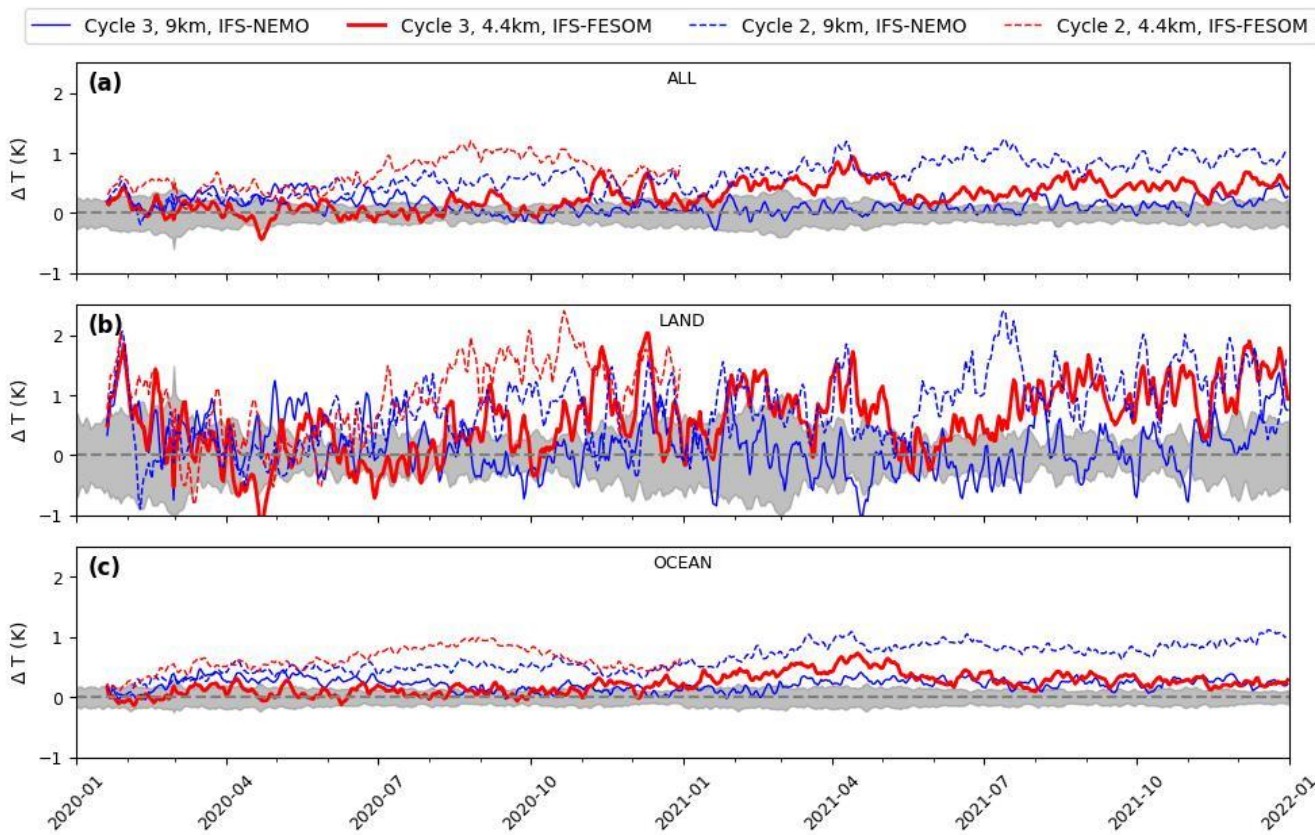

**Figure 5: Timeseries of 2-metre temperature global (a), only over land (b) and only over ocean (c) with respect to ERA5, for the years 2020-2021**. The shaded area shows the ERA5 standard deviation between 2012-2021. The evolution of the 2-metre temperature over 5 years is shown in Figure C1 in Appendix C.

### 3.1.4 Improved precipitation characteristics in Cycle 3 vs Cycle 2 and larger-scale impacts

Snapshots of cloudy brightness temperature and precipitation over the Indian Ocean (Fig. 6) illustrate that after 12 days of simulation in Cycle 3, there are biases in the characteristics of precipitating deep convection compared to satellite observations, even after the developments for Cycle 3 (see Section 3.1.1) were introduced. The observations show multiple mesoscale convective systems (MCS), which are associated with strong precipitation intensities and large anvil clouds. Neither the baseline 9 km Cycle 3 simulation nor the 4.4 km simulation manage to represent the MCS as observed. At 9 km, the convective cells are not well defined with wide-spread areas of weak precipitation. Indeed, precipitation intensity is underestimated in this setup, with precipitation intensity rarely exceeding 10 mm/hour (Fig. 7a). Instead of organising into MCS, hints of spurious gravity waves initiated from parametrized convective cells can be seen in the precipitation snapshot, emanating in different directions.

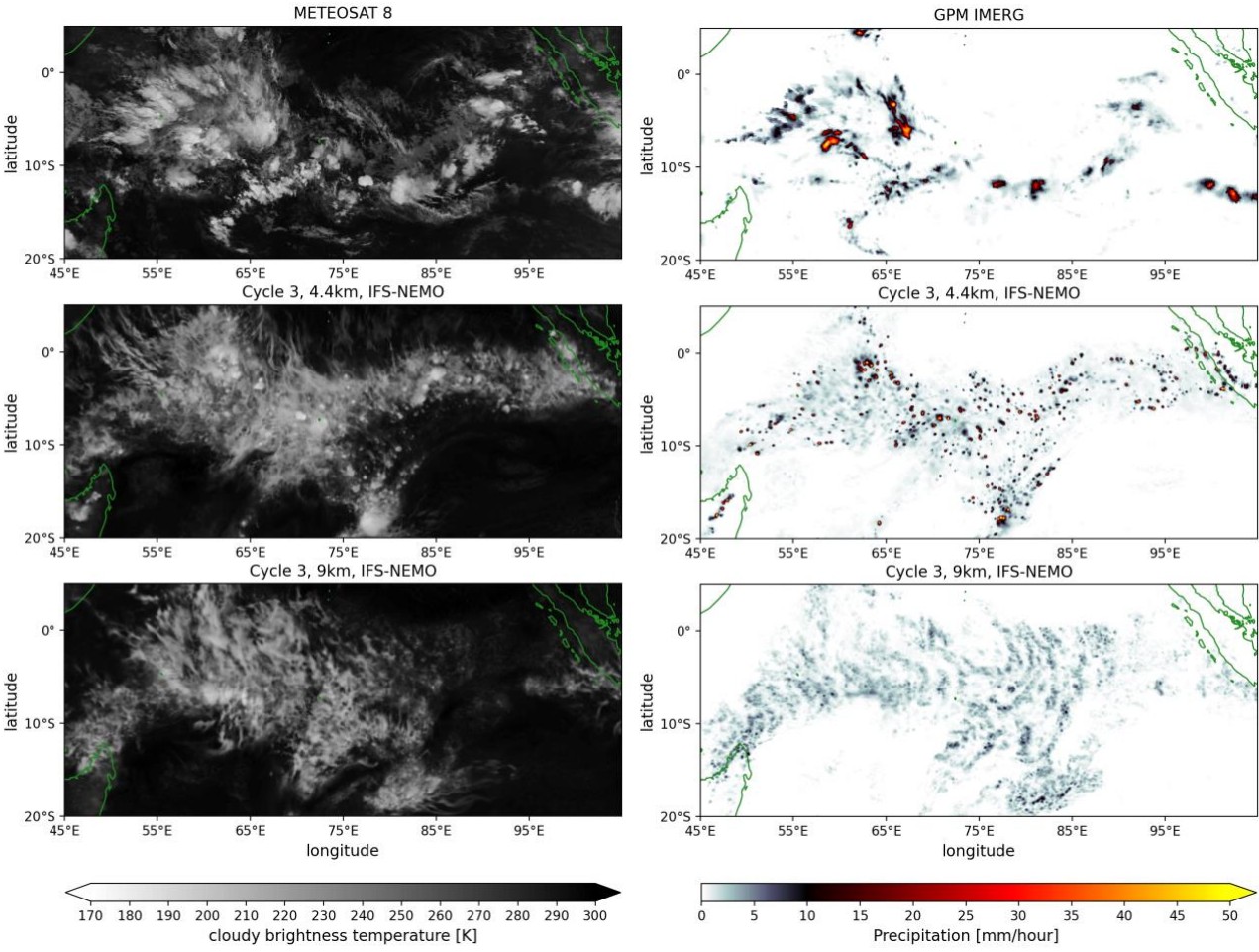

**Figure 6: Snapshot for 31/01/2020 at 21:00 UTC of infra-red brightness temperature (left) and hourly precipitation rate (right) over the Indian Ocean,** from observations (Meteosat 8 SEVIRI channel 9 and GPM IMERG, 1st row), and at forecast day 12 of IFS-NEMO 4.4 km (2nd row) and 9 km (3rd row) simulations. The simulations use the nextGEMS Cycle 3 setup except that they are run with a satellite image simulator and are coupled to NEMO V3.4 (ORCA025).

However, at 4.4 km resolution, the deep convection scheme is much less active, as the cloud base mass flux has been reduced by a factor of 6 compared to its value at 9 km (see Section 3.1.1). This setup features more realistic precipitation intensities, and particularly the strong precipitation of more than 10 mm/hour is close to the satellite retrieval GPM IMERG (Figure 7a), while the Cycle 2 simulations with Deep Off overestimate and with Deep On underestimate intense precipitation. In contrast, weak precipitation of 0.1 to 1 mm/hour is most strongly overestimated at 4.4 km resolution in Cycle 3. This is mostly precipitation that stems from the weakly active convection scheme. Solutions of how to reduce this drizzle bias are being worked on, e.g., through an increase of the rain evaporation rate.





A related issue is that the size of convective cells is too small, as illustrated by the size distribution of connected grid cells with
precipitation exceeding 3 mm/hour (Fig. 7b). The average size of a precipitation cell is rather similar in all simulations, and
only about half the value as in GPM IMERG. While GPM IMERG has a substantial number of precipitation cells that exceed
a size of $10^3$ grid points, which for example would correspond to a precipitation object of 5°x2°, this size is almost never
reached in the IFS simulations. The baseline simulations reach this size more often than the higher-resolution simulations, but
mainly in association with the spurious gravity waves, not because an MCS would be correctly represented. In summary, the
representation of intense precipitation has been improved from Cycle 2 to Cycle 3, but that has not led to more realistic
precipitation cell sizes. Even though it is possible that GPM IMERG overestimates precipitation cell size, cloudy brightness
temperature shows the same issue (Fig. 6). Work with other models (e.g., ICON, NICAM, SCREAM) has also shown that an
underestimation of precipitation cell size is a common issue in global km-scale resolution simulations, in some models even
leading to "popcorn" convection, and will require more attention in the future.

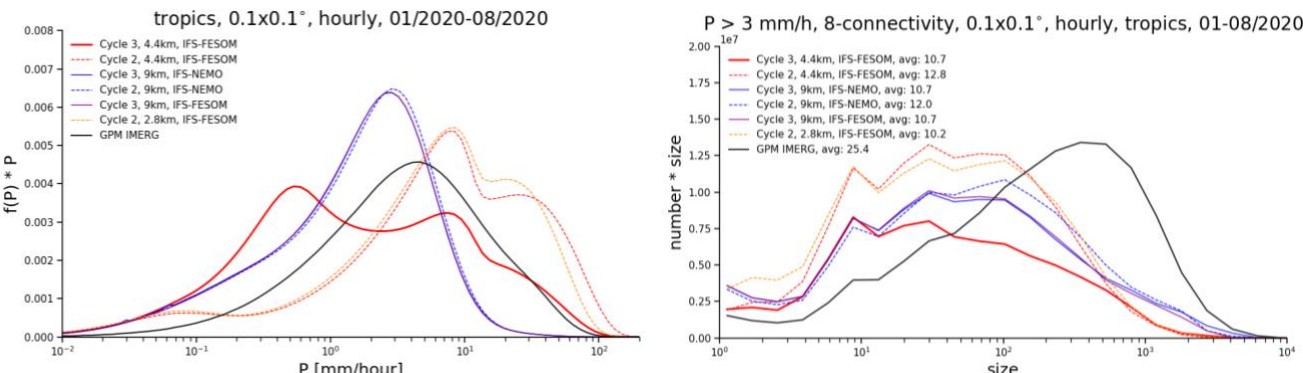


**Figure 7: (a) Frequency times bin intensity of hourly precipitation intensity in the tropics (30S-30N), conservatively**
**interpolated to a 0.1° grid from January to August 2020.** Following Berthou et al. (2019), the bins are exponential, meaning
that the area under the curve represents the contribution of that intensity range to the mean. (b) Histogram of precipitation cell
size times bin size, using a similar approach as in (a). The precipitation cell size is defined as the number of connected grid
cells on a 0.1° grid (also considering diagonal neighbours) where precipitation exceeds 3 mm/hour, counting cells in the whole
tropics (30°S-30°N), again from January to August 2020. The average precipitation cell size is given in the legend. The
observational estimate is from GPM IMERG.

As already mentioned in Section 3.1.1, the characteristics of meso-scale organisation of convection also affect the larger scales.
For example, in Cycle 2 simulations with Deep Off, the ITCZ often organises into a continuous and persistent line of deep
convection over the Pacific at 5°N (see Figure D1 in Appendix D) and as a consequence, the zonal mean precipitation is
strongly overestimated. This bias improved significantly from Cycle 2 to Cycle 3, when switching from a setup with no deep
convection scheme in Cycle 2 (at 2.8 and 4.4km resolution) to a setup with reduced cloud base mass flux in Cycle 3 (at 4.4km).





While the peak of precipitation around 5°N was overestimated by a factor of 2 during individual winter months in the 2.8 and
4.4km Cycle 2 run (see Figure D2 in Appendix D), the 4.4km Cycle 3 run shows a much reduced bias, and the peak at 5°N is
thus perfectly aligned with the GPM IMERG observations during September-December (Figure 8d). The 9 km baseline run
did not change significantly from Cycle 2 to Cycle 3 but it also shows some small improvements with regards to the
overestimation of the precipitation peak at 5°N.
Comparing the FESOM and NEMO runs, it is striking that all FESOM runs overestimate precipitation in the Southern
Hemisphere tropics around 10°S, hinting at a biased large-scale circulation, while NEMO runs show some good agreement
with observations. The different seasons (Figure 8b-d) show an overestimation of precipitation at 10°S only during January-
April in the NEMO runs, while FESOM runs overestimate precipitation at 10°S during most of the year. Additionally, the
FESOM runs also slightly underestimate precipitation at the equator (particularly during January-April), hinting at a double
ITCZ bias, which is a common issue in coupled simulations at km-scale resolutions during boreal winter, e.g. in ICON
(Hohenegger et al., 2023). Compared to ICON and other global coupled km-scale models that contributed to the DYAMOND
model intercomparison project (Stevens et al., 2019), the zonal mean precipitation biases in IFS nextGEMS Cycle 3 are of
similar nature and in part smaller than in the other models.

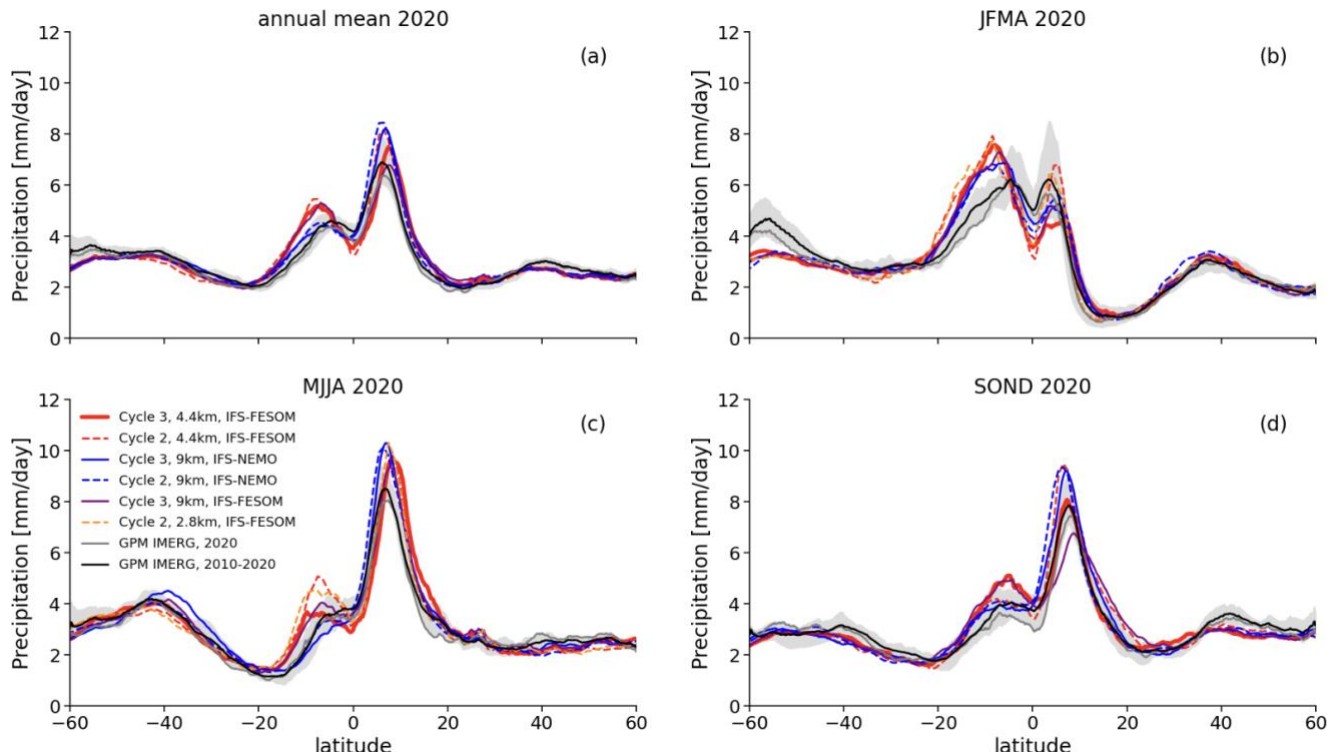


**Figure 8: Zonal-mean precipitation in nextGEMS Cycle 2 and 3, averaged over the year 2020 (a) and for different 4-**
**months periods in 2020, January-April (b), May-August (c) and September-December (d).** Observations are from GPM



IMERG for the year 2020 and for the 2010-2020 climatological period, indicating the climatological range of individual years
via the grey shading.

**3.1.5 Stratospheric Quasi-Biennial Oscillation**

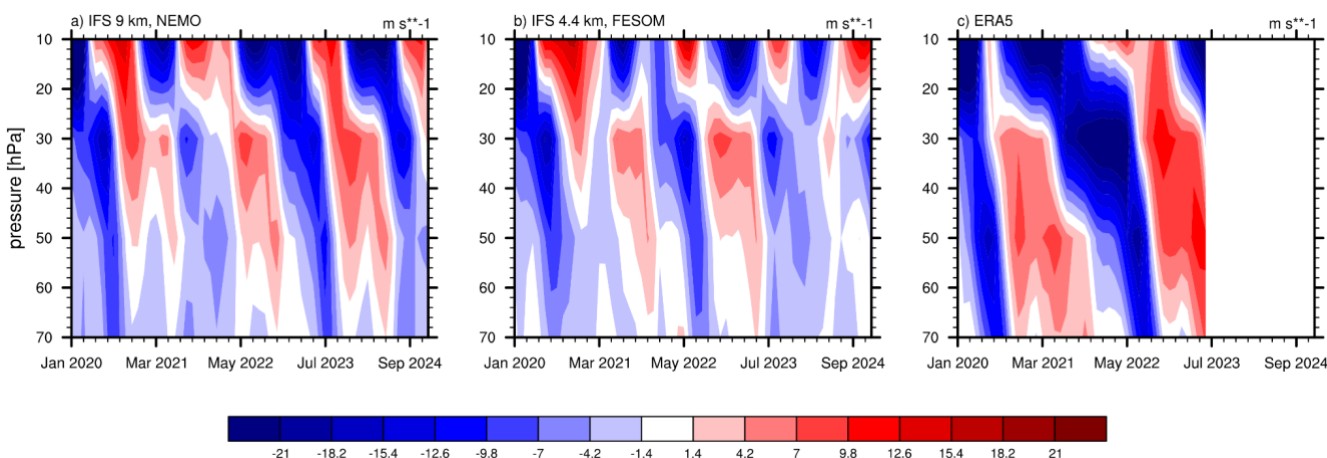


**Figure 9: Time evolution of monthly-mean zonal winds, averaged over the equatorial band 10S-10N,** for a) the IFS 9 km
Cycle 3 simulation with NEMO, b) the IFS 4.4 km Cycle 3 simulation with FESOM, and c) the ERA5 reanalysis for reference.
The Quasi-Biennial Oscillation (QBO) in the equatorial stratospheric winds is driven by momentum deposited by breaking
small-scale convectively generated gravity waves (GWs) and large-scale Kelvin and Rossby-gravity waves (e.g., Baldwin et
al., 2001). The QBO can have a downward influence on the troposphere (e.g., Scaife et al., 2022) and it is thus important to
simulate it well in seasonal and decadal prediction models. As km-scale models explicitly resolve GWs to a large extent, they
have a potential to better simulate the QBO than lower resolution models (e.g., CMIP), which fully rely on GW
parametrizations. However, GW parametrizations are often tuned to get a good QBO in lower resolution models (Garfinkel et
al., 2022; Stockdale et al., 2022) and at higher resolution the resolved GW forcing can be overestimated with less freedom for
tuning. For example, whether parametrized deep convection is switched on or off has a large impact on resolved GWs, with
fully resolved convection generating more than two times stronger GW forcing (Stephan et al., 2019; Polichtchouk et al., 2021)
and a QBO period that is – as a result – too fast.
We find that the QBO is reasonably well simulated in the nextGEMS Cycle 3 simulations at 9 km and even at km-scale (4.4
km) resolution (Fig. 9). The periodicity is reasonable, peaking at around 20 months at 30hPa for both simulations (calculated
by performing FFT on the monthly timeseries). This can be probably further improved by tuning the strength of parametrized
non-orographic GW drag, which is still on with reduced magnitude in both 9km and 4.4km simulations, reduced to 70% and
35%, respectively, compared to that at 28 km resolution.





In the lower stratosphere below 40hPa, the amplitude of the QBO, however, is underestimated (compare panels a-b) to panel
c) in Fig. 9), especially for the eastward phase. This deficiency is also observed in many lower-resolution models (Bushell et
al., 2022). We hypothesise that the overall reasonable QBO simulation at km-scale resolution might partly be due to the
parametrization for deep convection being still "slightly on" in the Cycle 3 simulations with IFS, as detailed in the previous
section.
**3.2 Ocean, Sea ice, and Waves**
**3.2.1 Key issues and model developments**
From a model development point of view, one of the main purposes of the nextGEMS Cycle 3 simulations was to set up and
test a fully-coupled global model that runs over multiple years and still does not show drift in global mean surface temperature
and other main climate characteristics, prior to performing the final multi-decadal integrations foreseen in nextGEMS. To
reduce drift (Figure 5), in particular over the Southern Ocean where the model in Cycle 2 had shown a strong warming over
the ocean with time compared to the ERA5 range for 2020-2021, the FESOM ocean component has been updated to the latest
release version 2.5 and coupling between the ocean and atmosphere has been improved.
**Warm biases over the ocean**
The warming ocean in Cycle 2 leads to an overall warming of the atmosphere as well. The 4.4km IFS-FESOM simulations in
Cycle 2 with 5km resolution in the ocean had shown a warming over the Southern Ocean in winter and year-round in the
tropics. For Cycle 3, the latter has been significantly improved by tuning the TOA balance and by using partially active
parametrized convection, while the former has been solved by a combination of different factors, namely (i) improvements in
the consistency of the heat flux treatment between the atmosphere and ocean/sea ice component, (ii) heat is taken from the
ocean in order to melt snow falling into the ocean, which had been overlooked before, (iii) the activation of a climatological
runoff/meltwater flux around Antarctica (COREv2, Large and Yeager 2009), and (iv) a general update from FESOM2.1 to
FESOM2.5 (Rackow et al. 2023c, https://github.com/FESOM/fesom2/releases/tag/2.5/). The resulting more realistic
temperature evolution in Cycle 3 is discussed in Section 3.2.2.
**Sea ice performance**
In Cycle 1 and 2, the sea ice representation in IFS-FESOM showed prominent deviations from the observed seasonal cycle in
the Ocean and Sea Ice Satellite Application Facility (OSI-SAF) dataset. This could be addressed mainly by correcting the
shortwave flux over ice with the release of FESOM version 2.5. The resulting sea ice performance in Cycle 3 is discussed in
Section 3.2.3.



### 3.2.2 Improved Southern Ocean temperature evolution

As already mentioned in section 3.1.3, IFS-FESOM simulations in Cycle 2 (TCo2559 and NG5 grid in the ocean) had shown a warming over the Southern Ocean in winter and year-round in the tropics. For Cycle 3, the improvement in IFS-FESOM 4.4km is particularly evident when comparing to the operational 9km IFS setup with NEMO V3.4. While the Southern Ocean shows a similar magnitude of anomalies in IFS-FESOM TCo2559-NG5 in year 5 compared to the first year, there appears to be an increase of anomalies over time in IFS-NEMO (Figure 10). This has been confirmed in a second set of IFS-FESOM simulations at TCo399 resolution (28km), and on the tORCA025 ocean grid (not shown).

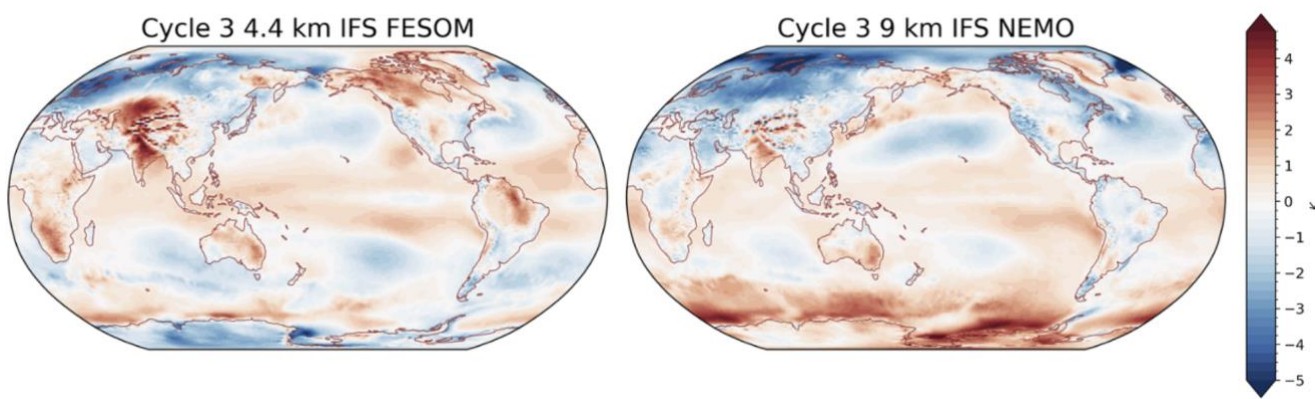

**Figure 10: Anomaly of annual-mean 2m temperature in year 5 of the nextGEMS Cycle 3 simulations, initialised on 20 January 2020, compared to the mean of ERA5 over 2020-2021**. (left) IFS-FESOM 4.4km/NG5, and (right) IFS-NEMO 9km/ORCA025.

### 3.2.3 Integrated sea ice performance metrics

The performance of the nextGEMS Cycle 3 simulations is analysed in terms of the sea ice extent and sea ice edge position (Fig. 11). The Integrated Ice Edge Error (IIEE), the Absolute Extent Error (AEE), and the Sea Ice Extent (SIE) metrics are used for comparing the model simulations and daily 2020 remote-sensing sea ice concentration observations from the Ocean and Sea Ice Satellite Application Facility (OSI SAF). Specifically, the recently released Global Sea Ice Concentration climate data record (SMMR/SSMI/SSMIS), release 3 (OSI-450-a; OSI SAF 2022) is considered in our analysis. The IIEE is a positively defined metric introduced by Goessling et al. (2016), and it is commonly used for evaluating the correctness of the sea ice edge position in Arctic and Antarctic sea ice predictions (Zampieri et al. 2018, Zampieri et al. 2019). We compute the IIEE by summing the areas where the model overestimates and underestimates the observed sea ice edge, here defined by the 15% sea ice concentration contour. The SIE is the hemispherically integrated area where the sea ice concentration is larger





than 15%. Finally, the AEE represents the absolute difference in the hemispheric SIE of models and observations, therefore
not accounting for errors arising from a different distribution of the ice edge in the two sets.




**Figure 11: (a) Arctic daily Integrated Ice Edge Error (IIEE; solid lines) and Absolute Extent Error (AEE; dashed lines)**
**for three different Cycle 3 simulations. (b) is the same as (a), but for Antarctic sea ice.** The IIEE and AEE metrics are
computed by comparing the three model runs against remote-sensing sea ice concentration observations from OSI-SAF. (c)
and (d) show the Arctic and Antarctic sea ice extent for two different Cycle 3 simulations from 2020 until the end of 2024.

All model configurations show substantial errors in representing the initial state. In the Arctic, the error grows in the first
simulation days in response to the active coupling between the sea ice components and the IFS atmospheric model (Fig. 11a).
In the Antarctic, an initial error growth takes place for the IFS-NEMO model configuration, while modest error mitigation is
seen for the two IFS-FESOM configurations (Fig. 11b). The latter feature suggests that a coupled setup could be better suited
to represent the Antarctic sea ice processes in the FESOM models, at least for this specific instance. Both in the Arctic and
Antarctic, the initial error of the IFS-NEMO configuration is substantially lower than that of the IFS-FESOM configurations.



This behaviour is expected since NEMO performs active data assimilation, while the sea ice in FESOM is only constrained by
the ERA5 atmospheric forcing (Hersbach et al. 2020) imposed during the ocean-sea ice model spinup. In the Antarctic, the
initial error differences diminish quickly and, after a couple of months, the errors of IFS-NEMO and IFS-FESOM are similar.
In the Arctic, IFS-NEMO exhibits residual prediction skill over IFS-FESOM in late spring, four to six months after the
initialization, possibly due to a more accurate description of the Arctic Ocean heat content influenced by the use of proper
ocean data assimilation techniques. After the initialization, the pan-hemispheric sea ice model performance is similar for the
three configurations, and attributing the error differences to the use of different model resolution or complexity is not obvious,
confirming previous findings (e.g., Streffing et al. 2022; Selivanova et al., 2023). Overall, the model errors for the first year of
simulations are in line with state-of-the-art seasonal prediction systems (Johnson et al. 2019; Mu et al. 2020; Mu et al. 2022),
showing similar features in terms of seasonal error growth.

When considering longer timescales (5-year simulations), model drifts are visible for the IFS-NEMO configuration and, to a
lesser extent, for the IFS-FESOM setup. In particular, the NEMO setup appears to progressively lose the winter sea ice cover
in the Southern Ocean (Fig. 11d). This behaviour is not compatible with the observed interannual variability of the Antarctic
sea ice and it is likely due to the near-surface temperature warming, which is not affecting the IFS-FESOM setup. Our
hypothesis is that the initialisation strategy for FESOM and NEMO accounts for part of the discrepancies in the multi-year
drift between IFS-NEMO and IFS-FESOM. We found that active data assimilation improved the model performance for the
initial months, while an uncoupled ocean spinup might be preferable for minimizing the drift towards the ocean model's
equilibrium state during the 5-year coupled simulation. In the Arctic, the sea ice extent tends to increase progressively in both
the FESOM and NEMO setups, with an additional dampening of the seasonal cycle observed for NEMO (Fig. 11c). Different
multi-year drift regimes between NEMO and FESOM could be also attributed to diverse complexity of the underlying sea ice
models. The more sophisticated physical parametrizations of the NEMO V3.4 configuration could respond more to the active
coupling with IFS compared to the FESOM setups.

### 3.2.4 Wind and waves

As written above in Section 2, in the IFS there is an active two-way coupling between the atmosphere and ocean waves. Surface
wind stress generates ocean surface waves and in turn those waves modulate the wind stress. The increase in resolution from
4.4km relative to the 9km for the IFS-FESOM simulations results in significant increases in wind speed in the storm tracks
(~50S and ~45N; Fig. 12a), most likely due to the increased ability to resolve the intense winds in the extratropical cyclones.
This increased resolution looks to be particularly important for the Southern Ocean, as the 4.4km simulation is the only one of
the three simulations that can achieve winds of realistic intensity in this area. We also note a significant improvement in the
trade winds (~15N) for the 4.4km IFS-FESOM simulation.

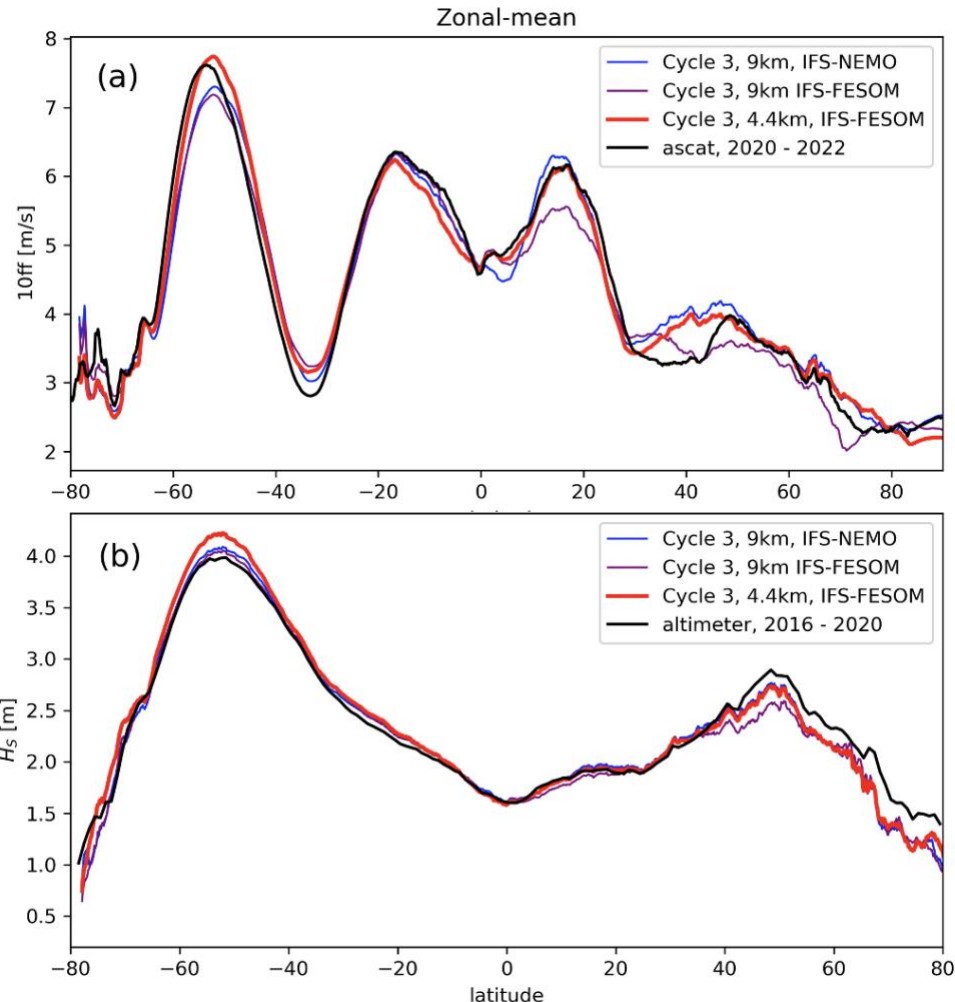

**Figure 12: Zonal-means of 10-metre wind speed '10ff' over ocean (a) and significant wave height (b) in nextGEMS Cycle 3.** Observations in black are from Copernicus Marine Service for wind speed ('ascat'; scatterometer combined with ERA5), and the ESA-CCI (v3) cross-calibrated altimeter record for wave height ('altimeter').

The waves in the storm tracks are also significantly larger (Fig. 12b). The increased wind is likely partly responsible for this increase. The second factor likely playing a role here is the change in fetch, i.e. the area of ocean over which the wind is contributing to wave growth. A notable decrease in mean sea ice concentration (more than ten percent) takes place in the 4.4km simulation (Fig. E1a), thereby freeing up the ocean surface here for wave growth. These changes can be directly seen in the wave field in the according areas (Fig. E1b). These waves then continue to grow with the wind as they propagate into the Southern Ocean, thereby contributing to the larger waves seen in this region. For the NH storm track, this points to an





639 improvement with respect to altimeter observations, but for the Southern Ocean the 4.4km simulation is now somewhat

640 overestimating the waves.

**3.3 Land**

**3.3.1 Key issues and model developments**

Performing simulations at the km-scale inherently brings a richer picture in the atmosphere and ocean in terms of small-scale features, as more scales become explicitly resolved. To gain the full benefit of the resolution over land, it is important that the surface information is also at an equivalent or finer resolution. Therefore, work at ECMWF in recent years has been directed to provide the IFS surface model ECLand (Bousetta et al., 2021) with surface global ancillary information of a resolution down to 1 km or finer, and to include additional processes that become relevant at those scales. These developments had always the improvement of the operational IFS as a goal and focused, therefore, on timescales from days to a few months. nextGEMS simulations present a timely opportunity to test these changes in parallel before they become operational, and to assess their impact when fully coupled on multi-annual timescales. Most of the developments in this section are described in more detail by Bousetta et al. (2021). Here in this section, nextGEMS Cycle 2 and Cycle 3 will refer to IFS CY48r1 (ECMWF, 2023b) and CY49r1 (scheduled for 2024), respectively.

**Km-scale surface information**

An improved land-water mask was included for nextGEMS Cycle 2. The original source belonging to the Joint Research Centre (JRC) had a nominal resolution of 30m. The mask was further improved by including glacier data and new land-water and lake fraction masks. In parallel, lake depth data was improved (Bousetta et al., 2021).

Further changes to the land-water mask were tested in nextGEMS Cycle 3. The Land Use/Land Cover maps (LU/LC) used before nextGEMS Cycle 3 were based on those from GLCCv1.2 data (Loveland et al., 2000), which is based on observations from the Advanced Very High Resolution Radiometer (AVHRR) covering the period 1992–1993. They had a nominal resolution of about 1km. In nextGEMS Cycle 3, we used new maps, based on ESA-CCI, which exploit the high resolution of recent remote sensing products down to 300m and will pave the way to enable observation-based dynamic LU/LC maps in the future. These maps lead to a more realistic overall increase of low vegetation cover compared to the GLCCv1.2-based maps, at the expense of the high vegetation cover. The new conversion from ESA-CCI to the Biosphere-Atmosphere Transfer Scheme (BATS) vegetation types used by ECLand also reduces the presence of ambiguous vegetation types like 'interrupted forest' or 'mixed forest'. In addition, work has been done on upgrading the Leaf Area Index (LAI) seasonality and its disaggregation into low and high-vegetation LAI. This improves, among others, the previously found overestimation of total LAI during March-April-May (MAM) and September-October-November (SON). This revised description of the vegetation will also be



used in the next operational IFS cycle (49R1), and an initial implementation and evaluation is presented in Nogueira et al
673  (2021).
The thermodynamic effects of urban environments emerge at the surface as models refine resolution down to the km-scale and
the rural-urban contrast sharpens. To determine where to activate the urban processes at the surface, a global map of urban
land cover is used here in our nextGEMS Cycle 3 simulations. This map, based on information provided by ECOCLIMAP-
SG at an initial 300m horizontal resolution (McNorton et al., 2023; Faroux et al., 2013), will also be used in the next operational
IFS cycle (49R1).

**Km-scale surface processes**

The presence of the fine spatial information described above opens the path to simulate relevant km-scale processes and
interactions. In particular, the representation of snow, 2-metre temperature, and urban areas was improved as detailed in the
following.
A newly developed multi-layer snow scheme was implemented in IFS CY48r1 and was already used in the nextGEMS Cycle
2 (Arduini et al. 2019), substituting the existing snow bulk-layer scheme. The new scheme dynamically varies the number of
snow model layers depending on the snow depth and provides snow temperature, density, liquid water content and albedo as
prognostic variables. In addition, snow and frozen soil parameters were modified for improved river discharge (Zsoter et al.,
2022) and permafrost extent (Cao et al. 2022). An additional upgrade in nextGEMS Cycle 3 was a package of changes to
ECLand which will be included in the next operational IFS cycle (49R1). This contains an improved postprocessing of 2-metre
temperature reducing the warm bias present occasionally under very stable conditions. It also contains a significant upgrade
to the representation of the near-surface impact of urbanized areas. For this purpose, the urban scheme developed in ECLand
was activated. This scheme considers the urban environment as an interface connecting the sub-surface soil and the atmosphere
above (McNorton 2021, McNorton 2023). The urban tile comprises both a canyon and roof fraction. In terms of energy and
moisture storage, the uppermost soil layer is not specific to the tile but represents a grid-cell average. This results in a weighted
average that accounts for both urban and non-urban environments. The albedo and emissivity values used in radiation exchange
computations (McNorton 2021, McNorton 2023) are determined based on an assumption of an "infinite canyon," taking into
account "shadowing." The roughness length for momentum and heat follows the model proposed by Macdonald et al. (1998)
and varies according to urban morphology. Simplified assumptions regarding snow clearing and run-off are incorporated based
on literature estimates (e.g., Paul & Meyer, 2001). Illustrative examples of urban cover characteristics and the impact of
accounting for urbanised areas in Cycle 3 vs Cycle 2 simulations are highlighted in Section 4.3.





## 4 Selected examples of significant advances in km-scale nextGEMS simulations

In this section, we will highlight three examples of notable advances in the Cycle 3 4.4km nextGEMS simulations that emerge due to the km-scale character of our simulations. Besides successes in the representation of the Madden-Julian Oscillation (MJO), an important variability pattern that is linked to the monsoons, we also provide examples of small-scale air-sea ice interactions in the Arctic, and touch on atmospheric impacts due to the new addition of km-scale cities in the IFS. We expect more in-depth process studies as part of ongoing analyses within the nextGEMS community and as part of dedicated future work.

### 4.1 MJO propagation and spectral characteristics of tropical convection

The MJO is a dominant intraseasonal variability mode in the tropics, characterised by slow eastward propagation of large-scale convective envelopes over the Indo–Pacific warm pool (Madden and Julian, 1972). The MJO convection and circulations have profound impacts on weather and climate variability globally (Zhang, 2013), so that it is important to reproduce the MJO in global circulation models (GCMs) targeting seasonal-to-decadal simulations. Having the MJO well represented in models is indicative of a better tropical or global circulation. Because the reproducibility of the MJO is highly sensitive to the treatment of cumulus convection (e.g., Hannah and Maloney, 2011), many conventional GCMs that adopt cumulus parametrizations, which have uncertainties in the estimation of cumulus mass fluxes and moistening and heating rates, still struggle with simulating important MJO characteristics such as amplitudes, propagation speeds, and occurrence frequencies appropriately (e.g., Ling et al., 2019; Ahn et al., 2020 Chen et al., 2021). This issue might be improved by km-scale simulations as a result of more accurate representation of moist processes, as represented by the first success of an MJO hindcast simulation with NICAM (Miura et al. 2007), but also other physical processes (besides convection) play a role for skilful MJO simulations (Yano and Wedi, 2021).





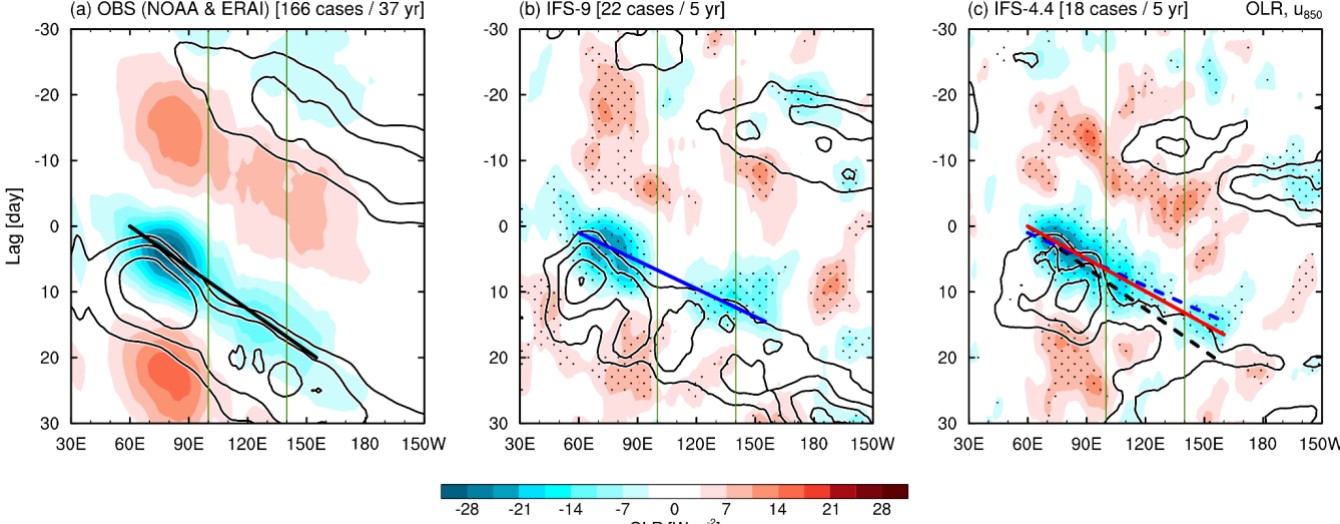

**Figure 13: Propagation characteristics of MJO convection and circulations composited from (a) observations and (b) IFS 9km simulation with NEMO and (c) IFS 4.4km simulation with FESOM.** Time–longitude diagrams of lagged-composite intraseasonal OLR (shading) and 850-hPa westerly wind anomalies (contours) averaged over 10°N–10°S. Contour interval is 0.5 m/s, with zero contours omitted. Stippling in (b) and (c) denotes statistical significance of OLR anomalies at the 90% level (All shading in (a) satisfies this significance). The number of detected MJO cases is denoted at the top of the figures together with analysis periods. Green lines indicate the longitudinal range over the Maritime Continent, and black, blue, and red lines indicate the centre of MJO convective envelopes for the observations, the 9 km simulation, and for the 4.4 km simulation, respectively.

Figure 13 illustrates the MJO propagation characteristics in the Cycle 3 4.4km IFS-FESOM simulation in comparison with the observations and the 9km IFS-NEMO simulation, using the MJO event-based detection method (Suematsu and Miura, 2018; Takasuka and Satoh, 2020). Note that the observational reference is made by the interpolated daily OLR from the NOAA polar-orbiting satellite (Liebmann and Smith, 1996) and ERA-Interim reanalysis (Dee et al. 2011) during the period of 1982–2018. While the 9km simulation already does a very good job and both the 9km and 4.4km simulations can reproduce the overall eastward propagation of MJO convection coupled with zonal winds (Figures 13b and 13c), the 4.4km simulation allows to improve even further in terms of amplitudes and propagation speeds. Specifically, MJO convective envelopes in the 4.4km simulation are continuously organised when they propagate into the Maritime Continent (see OLR anomalies in 100°–120°E), and their propagation speeds become slower than in the 9km simulation and thus closer to those in the observation. We hypothesize that km-scale resolutions and partially resolved convection can better represent convective systems around complex land-sea distributions and topography. Nevertheless, the 4.4km simulation still retains several biases compared to the observed MJOs such as much faster propagation and weaker convection amplitudes to the east of 120°E (i.e., the eastern part of the Maritime Continent).





746

**Figure 14: Wavenumber-frequency power spectra of equatorially (a-c) symmetric and (d-f) antisymmetric components of tropical convection measured by OLR anomalies in (a, d) observations, (b, e) IFS 9km simulation with NEMO, and (c, f) IFS 4.4km simulation with FESOM.** Power spectra are summed from 15°S to 15°N, and plotted as the ratio of raw to background power. Abbreviations of WIG, TD, ER, MRG, and EIG indicate westward inertia-gravity waves, tropical depressions, equatorial Rossby waves, mixed Rossby-gravity waves, and eastward inertia-gravity waves, respectively. Dispersion curves for corresponding equatorial waves are plotted for equivalent depths *h* = 12, 25, and 50 m. *n* denotes the number of meridional modes.

754

Notwithstanding the intricacies of tropical mesoscale circulations (Stephan et al, 2021), we further compare with linear Fourier analysis the appearance of convectively coupled equatorial wave activities between the observation and 9km and 4.4km simulations (Figure 14), following the methodology of Takayabu (1994) and Wheeler and Kiladis (1999). Several previous studies also evaluated the representation of equatorial waves in IFS simulations (Dias et al., 2018; Bengtsson et al., 2019). For the equatorially symmetric components of tropical convection (Figures 14a–c), the IFS simulations at both resolutions can



simulate Kelvin waves separated from the MJO, whereas the amplitudes of equatorial Rossby waves and tropical depression-
type disturbances (i.e., westward-propagating systems in several-day periods) are somewhat underestimated especially in the
4.4km simulation. Meanwhile, the representation of the equatorially antisymmetric wave modes are significantly improved in
the 4.4km simulation; both $n = 0$ eastward inertia-gravity waves and mixed Rossby-gravity waves can be reproduced with
amplitudes as large as in the observation.
**4.2 Sea ice imprint on the atmosphere**

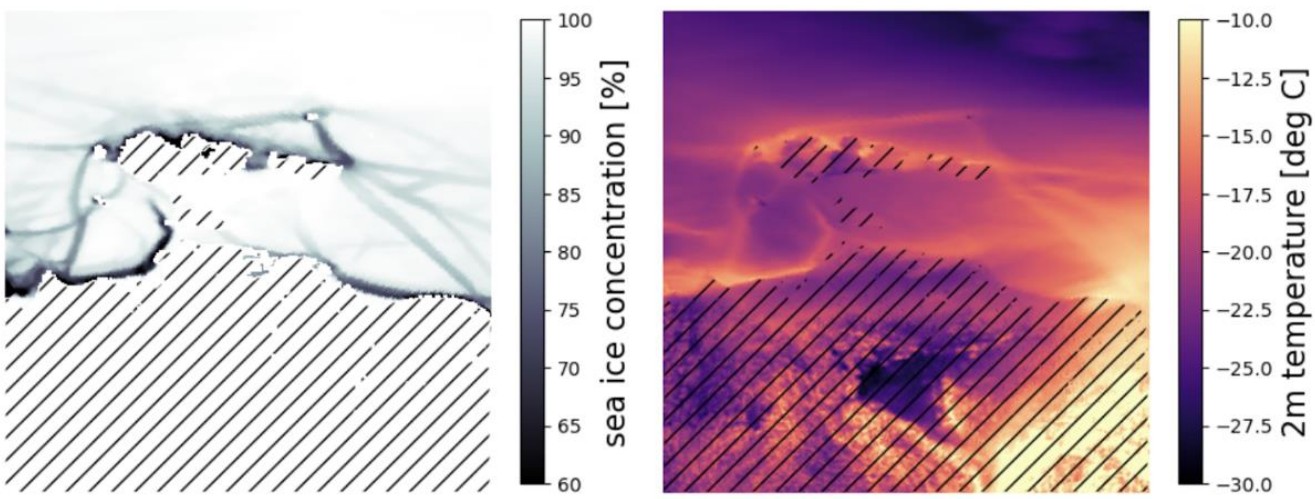


**Figure 15: Imprint of simulated Arctic sea ice leads on 2m-temperature in the Laptev Sea and East Siberian Sea.** (left)
Sea ice concentration field, (right) 2m-temperature field. The panels depict 13 February 2020, 08:00, in the IFS-FESOM Cycle
3 simulation with TCo2559 (4.4 km), coupled to the NG5 ocean (~4-5 km resolution in this area). Dashed lines represent land
areas.
Leads are narrow open areas in the sea ice cover that typically form after deformation events, such as caused by a persisting
Arctic storm over the ice cover. Individual leads can form typical 'linear' channels of several kilometres length, while the
larger connected lead systems can extend up to hundreds of kilometres (Overland et al. 1995) or even cross the entire Arctic.
They are detectable in satellite synthetic-aperture radar images (von Albedyll et al. 2023). Especially in winter, open leads can
significantly impact the stability of the atmospheric column and other atmospheric parameters above them. A change in sea
ice cover of 1% can cause near-surface temperature responses around 3.5 K (Lüpkes et al., 2008).
At the km-scale resolution employed here, there is first evidence of resolved linear kinematic features in the sea ice cover at a
grid-spacing of ~4-5km in our coupled simulations (ECMWF News Item, 2022). With resolutions of 4.4km and 2.8km, the
atmosphere can thus 'see' these narrow features in the sea ice cover and simulate a response explicitly. Similar to the effect





that meso-scale ocean eddies can have on the atmosphere above them (Frenger et al. 2013), we find that the leads in sea ice
can strongly modulate the atmospheric state above them in our simulations. To give an example from the Arctic winter, north
of Siberia in the Laptev and East Siberian Sea, due to the relatively warm ocean compared to the atmosphere, 2m-temperature
anomalies over sea ice leads can often reach 10–20K against the surrounding closed sea ice cover background (Figure 15).
While the realism with respect to the size, number, spatial distribution, and orientation of the simulated leads still needs to be
quantified (Hutter et al. 2022), the direct simulation of sea ice lead effects within a coupled km-scale climate model is entirely
novel and opens up new areas of research. Potential climate impacts of this air-ice-ocean interaction on the atmospheric
column, such as Arctic clouds (Saavedra Garfias et al., 2023), will be one focus of our future work.
**4.3 Cities and urban heat island effects**

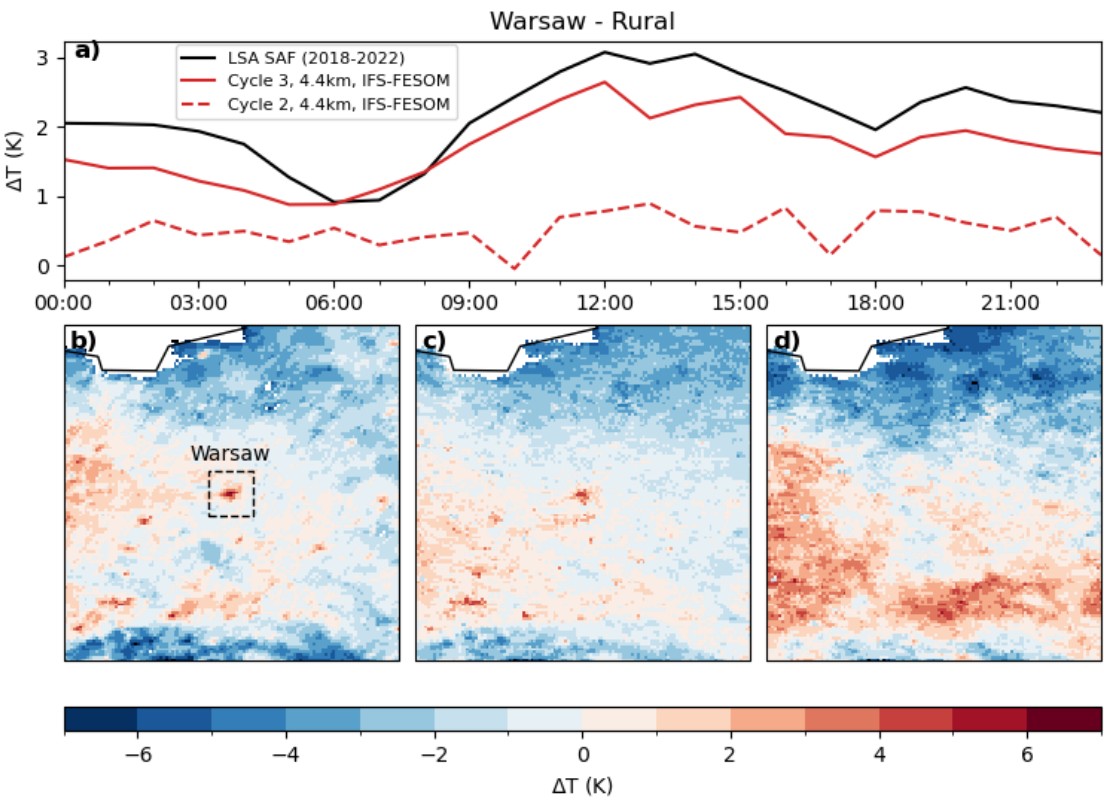



**Figure 16: Diurnal cycle of land surface temperature (LST) difference between the city of Warsaw and its rural**
**surroundings,** for a) the summer months (JJA) during clear-sky conditions (5-year mean). The IFS 4.4km simulations are
given with red lines (Cycle 2 dashed, Cycle 3 solid), observations from LSA SAF are given in black. The bottom panels show





JJA-mean clear-sky LST anomaly maps at 13:00 local time, with respect to the surrounding rural LST average, for b)
observations from LSA SAF, for c) IFS 4.4km Cycle 3 with urban scheme, and for d) IFS 4.4km in Cycle 2 without urban
scheme and using older land use/land cover maps.

Between Cycles 2 and 3, significant improvements have been achieved in representing urban heat island effects around the
globe at the km-scale (Fig. 16). To give an example, the difference in land surface temperature (LST) between the city of
Warsaw and its more rural surroundings during the 5-year clear-sky hours (in the JJA season) depicts a clear urban heat island
effect (Fig. 16, a), with temperature anomalies compared to the rural areas in exceedance of typically 1K over any given day,
and exceeding 2K around noon. When comparing with observations from the Satellite Application Facility on Land Surface
Analysis (LSA SAF) LST product (Trigo et al. 2008), the results in Cycle 3 show a closer fit to the satellite product than was
possible in Cycle 2; both the average temperature difference over the day, as well as its temporal variability, is better captured
(Fig. 16, a). Although the sub-diurnal variability is qualitatively well represented, the Cycle 3 modelled urban-rural contrast is
systematically around 0.5 K smaller than in observations. We hypothesise that missing anthropogenic heating as well as an
underestimation of the urban heat storage due to too low urban cover or building height may explain some of the discrepancies.
In terms of spatial variability of LST JJA-mean clear-sky anomalies, our Cycle 3 4.4km IFS simulation (year 2020) matches
km-scale details of the LSA SAF dataset (2018-2022) well (compare Fig. 16, b & c), while Cycle 2 4.4km IFS cannot provide
this local detail in the absence of updated land use/land cover maps plus urban scheme (Fig. 16, d). Note also that the changes
in high and low vegetation cover and vegetation types in Cycle 3 impact positively on the areas found to be too warm in Cycle
2 in the South and East of Warsaw. These results illustrate clearly that high-resolution surface information as well as an urban
scheme will be necessary in the context of the increasing need for local climate information on a city scale, and for local
projections of direct socio-economic relevance.

## 5 Summary and Conclusions

In this paper, storm- and eddy-resolving simulations performed with the nextGEMS configurations of the ECMWF Integrated
Forecasting System have been described and analysed. While we have also presented eddy-permitting simulations with IFS-
NEMO as the ECMWF operational baseline configuration, we have focused mostly on IFS-FESOM runs that feature not only
the highest atmospheric resolution (2.8km and 4.4km) but also an eddy-resolving ocean at 5km. The large-scale performance
in terms of the mean state has been presented, such as top-of-the atmosphere radiation balance and surface temperature biases,
but also important variability patterns (e.g. MJO and QBO) that can be analysed in 5-year long simulations. The illustrated set
of emerging advances in the km-scale nextGEMS simulations are first indications of the added value of km-scale modelling
and explicit simulation of smaller scales. We expect to be able to show more of these examples once longer simulations will
be available from the multi-decadal production simulations planned in nextGEMS for 2024. In this study it is the first time
that the model configuration and quality of the simulations with IFS-FESOM have been described; and it thus represents a



significant milestone both in terms of documenting this novel model capability and the scientific readiness of the coupled
modelling system.

A number of model developments along the nextGEMS model development cycles allowed to increase the realism of the km-
scale simulations. For example, activating mass fixers for water vapour, cloud liquid, ice, rain and snow made global water
non-conservation negligible and reduced energy non-conservation to an amount that is acceptable for long climate simulations,
even though further progress could be made by adding an energy fixer that corrects temperature. Importantly, global water
conservation turns out to be beneficial not only for long climate integrations, but also for the quality of ECMWF's medium-
range weather forecasts. Work for ECMWF's recent operational IFS upgrade in June 2023 (48r1) showed that the model
changes performed to fix the water and energy imbalances reduce the overestimation of mean precipitation at different
timescales and improve the skill scores for the recent operational resolution upgrade for medium-range ensemble weather
forecasts (ECMWF Newsletter 172, 2022). For example, the mean absolute error of precipitation against rain gauge
measurements is about 2–3% smaller in 9 km forecasts that ensure global water conservation compared to 9km forecasts
without water conservation. This is a great example of a model development from the nextGEMS multi-year simulations
feeding into the improvement of the operational NWP system at ECMWF.

Variability patterns that can be studied with the 5-year simulations performed so far in nextGEMS are the Madden-Julian
Oscillation (MJO) and the Quasi-Biennial Oscillation (QBO) in the equatorial stratospheric winds. The QBO is simulated with
reasonable periodicity, which is typically challenging for km-scale models without any active parametrization for deep
convection. The remaining shown deficiencies are likely due to the overly active vertical diffusion parametrization in stable
conditions, which will be addressed in an upcoming version of the IFS. The MJO is similarly well represented in both the 9km
and 4.4km simulations. In particular, however, MJO convective envelopes are continuously organised in the 4.4km simulation
when they propagate over the Maritime Continent, which is closer to observations. We think that this is not just an effect of
sampling different numbers of MJO events in our simulations and in the observations (simulated 5-year periods at 9km and
4.4km resolution versus long-term observational period) since the observed MJO for shorter periods of time (e.g., 2011-2015)
shows a similar result to the full observational record. The realistic representation of tropical variability and wave activity in
the IFS at 9km and 4.4km is the result of 15 years of sustained efforts in model developments, notably convection, cloud-
radiation interaction, and air-sea coupling (Bechtold et al. 2008, Dias et al 2018). The documented further improvements in
the 4.4km simulation compared to 9km can possibly result from reduced cloud base mass fluxes (i.e., more weight on explicit
convection), although the further detailed examination is left for our future work.

With our km-scale simulations that resolve mesoscale ocean eddies over large parts of the globe, we can also investigate
coupled effects between sea ice leads, open narrow channels in the sea ice cover, and the atmosphere above them for the first
time. Leads form during deformation events and can span over distances from several to hundreds of kilometres. From limited





observations and field campaigns it is known that sea ice leads can significantly impact the stability and temperature of the
atmospheric column, especially in winter. We find that our model can resolve the linear features of the leads and represent
explicitly the resulting heating of the atmosphere. This is a novel and promising approach that reveals new aspects of the air-
ice-ocean interaction.

The nextGEMS model configurations are also starting points for the Climate Adaptation Digital Twin in the Destination Earth
initiative, which aims to provide local climate information, for instance at the scale of cities, globally. The urban heat island
effect, which is the phenomenon of higher temperatures in urban areas compared to rural areas, is an aspect of socio-economic
importance that will need to be accurately represented by km-scale models in the future. In this study, we have shown that the
implementation of an urban scheme in the IFS for nextGEMS Cycle 3 can significantly improve the simulation of land surface
temperature (LST) over urban areas around the world, compared to previous model cycles that were missing specific urban
tiles. The example of Warsaw illustrates the improvement in both temporal and spatial variability of land surface temperatures
when compared to observations. We have also identified some limitations, such as nocturnal LST differences, which may be
related to the lack of some anthropogenic heating in the model. Our first results here demonstrate the necessity and benefit of
using an urban scheme in km-scale models for future efforts to provide reliable local climate information at the city scale.

While kilometre-scale model resolution is of benefit for the representation of the atmosphere, ocean, sea ice and land, it is also
of importance for our understanding of other components of the climate system that have not been covered in this study yet,
such as deep ocean circulation and ice sheet behaviour. For example, ocean heat transport at depth towards the Antarctic ice
sheet and ice-shelf cavities is localised in narrow canyons (Morrison et al. 2020). To resolve bathymetric features like this and
their potentially far-reaching impacts could be a strength of high-resolution models. Another example is the equilibration of
the Antarctic Circumpolar Current, which is a balance of the wind-driven circulation and the opposing eddy-induced
circulation cells. While transient ocean eddies can be parametrized to some degree, the effect of standing eddies (or meanders
of this current) are beyond what parametrizations can achieve (Bryan et al., 2014). First studies indicate that explicit simulation
of these effects with km-scale ocean models might be warranted to achieve higher confidence in projections of the Southern
Ocean and global sea level rise (van Westen and Dijkstra, 2021; Rackow et al. 2022).

We have demonstrated that kilometre-scale modelling, which will soon enable multi-decadal simulations, has become feasible
and offers advantages over lower-resolution models. The results presented here prove that our seamless model development
approach, where numerical weather prediction models are extended for km-scale multi-decadal climate applications, is useful
(Randall and Emanuel, 2024) and can benefit the original NWP application as well. As we have shown by running those
models for 5 years, the km-scale simulations improve the representation of atmospheric circulation and extreme precipitation,
but also enhance the coupling between the atmosphere, land, urban areas, ocean, and sea ice. We have revealed novel
interactions among these components for the first time that will be further explored in ongoing work. With upcoming multi-



decadal simulations from the nextGEMS and Destination Earth projects we will be able to generate even more statistics on
km-scale modelling soon, with an extended set of simulations from several models. These projects aim to provide accurate
and globally consistent information on local climate change - at the scales that matter for individual cities or local impact
modelling.

**Appendix A** - **Conservation properties of the IFS advection scheme and mass fixer approach**
The IFS uses a semi-Lagrangian (SL) advection scheme which is unconditionally stable and accurate and hence
computationally efficient. It is also multi-tracer efficient as many tracers can be transported with a relatively small overhead:
to advect a tracer or a prognostic variable (e.g. temperature, wind components), the upstream locations of model grid-points
must be computed (departure points) and the tracer must be interpolated at these locations for each advected variable (for
details, see Diamantakis and Váňa 2021). However, despite being accurate and efficient, as a tracer transport scheme it lacks
the property of conservation. In the absence of sources/sinks, the global mass of a tracer should remain constant, however, SL
advection changes slightly its global mass. This change depends strongly on the spatial characteristics of the tracer such as
smoothness of the field and its geographic location, with larger conservation errors for tracers that have sharp gradients and
interact with the orography.
Conservation properties are important for water and energy budgets, especially for high resolutions. A practical solution that
restores the global mass conservation of water tracers without altering the efficient and accurate numerical formulation of the
IFS, is the mass fixer approach. However, simple mass fixers which change each tracer gridpoint value by the same proportion
may result in unwanted biases in some regions. Hence, a more "local" approach is applied in the IFS advection scheme, which
was originally developed and tested for atmospheric composition tracers yielding accurate results when compared against
observations (Diamantakis and Fleming 2014, Diamantakis and Agusti-Panareda 2017). This is a "weighted" approach as the
correction of the tracer field at each grid point depends on a weight factor which is proportional to a local error measure. The
correction restores global conservation, using local criteria and it also preserves positive definiteness and monotonicity of the
field.



**Appendix B**

The 5km nextGEMS ocean grid in this study (termed 'NG5') makes use of the multi-resolution mesh capabilities provided by the FESOM ocean-sea ice model (Figure B1). From nextGEMS Cycle 2 and following cycles, FESOM was run with this new eddy-resolving ocean grid with spacing of less than ~5km (at the poles) and around 13km in the tropics. This grid, specifically designed by the Alfred Wegener Institute (AWI) to better match the high atmospheric resolution of 4.4km in the IFS, allows to better resolve areas of particular interest at higher resolution, such as the Western boundary currents or the Southern Ocean. The mesh was created with the JIGSAW-GEO package (Engwirda, 2017).

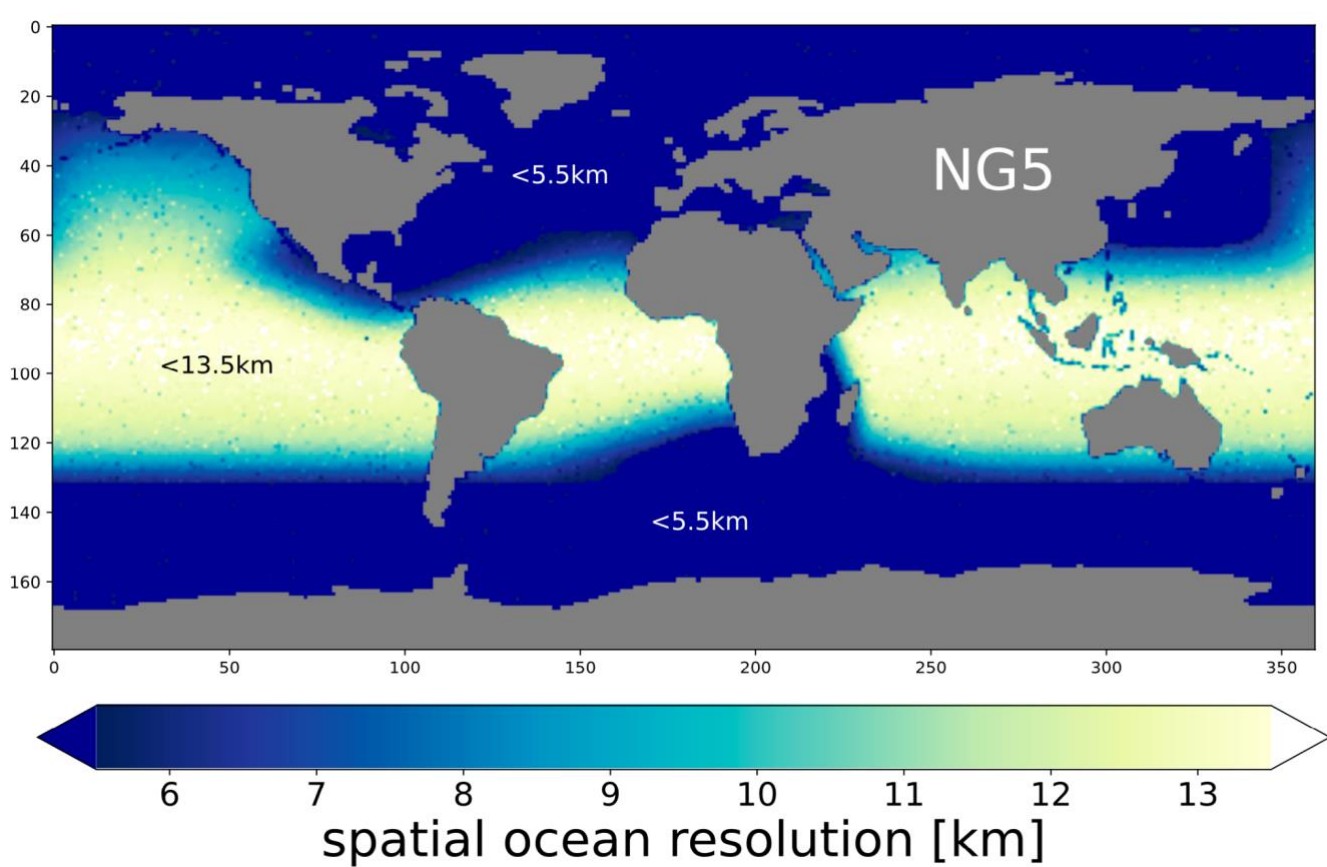

**Figure B1: Spatial ocean resolution in the nextGEMS 5km grid, NG5 [km].**



**Appendix C**

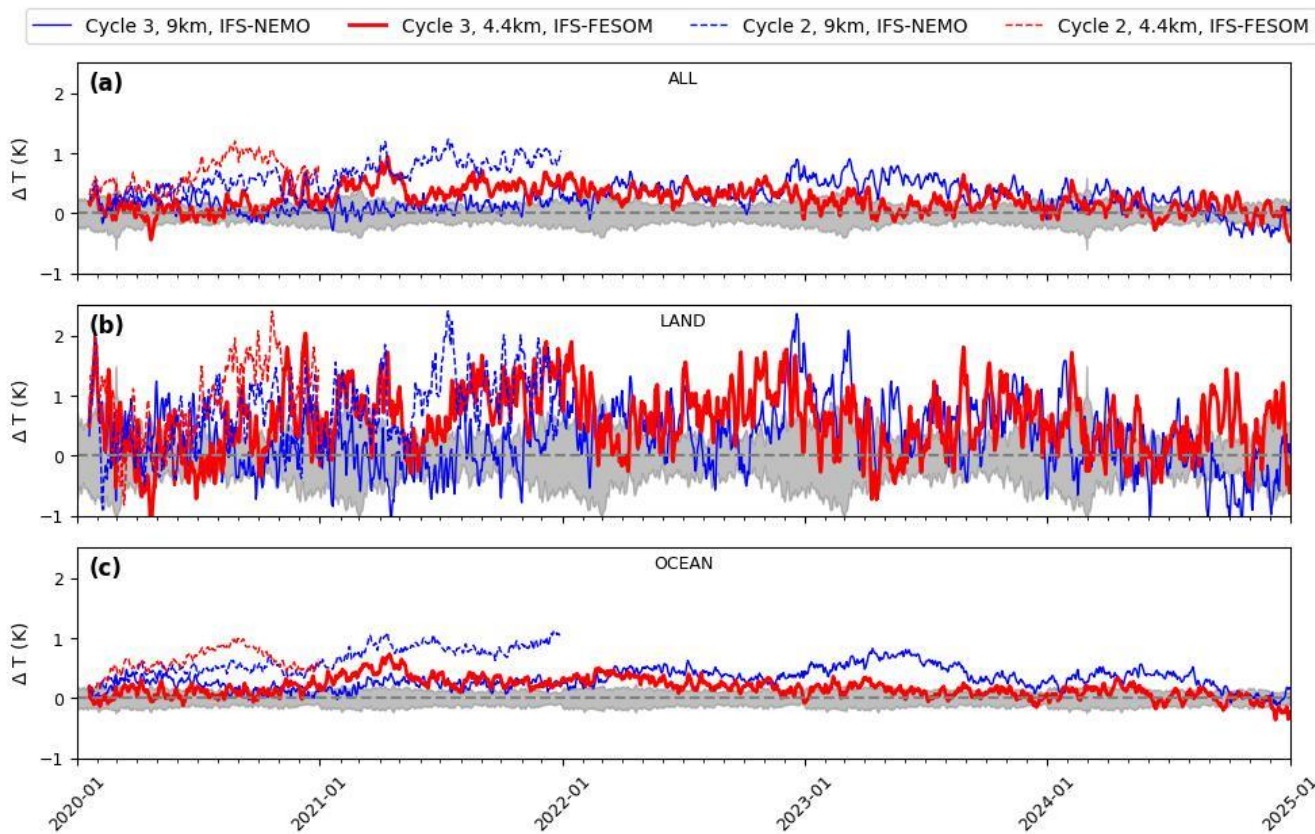

**Figure C1: Timeseries of mean 2-metre temperature in nextGEMS simulations for (a) global, (b) only over land, and**
**(c) only over ocean with respect to ERA5, for the years 2020-2024.** The shaded area shows the ERA5 standard deviation
between 2012-2021.




Appendix D



**Figure D1: Snapshots of precipitation in nextGEMS Cycle 2 and 3 simulations in the tropical Pacific at 5°N, compared to observations from GPM IMERG.** The ITCZ often organises into a continuous and persistent line of deep convection over the Pacific at 5°N in Cycle 2 at 4.4km and 2.8km resolution (lower two panels), with strongly overestimated zonal mean





precipitation along this latitude. In Cycle 3 this has been addressed via a reduced cloud-base mass flux with 4.4km resolution.
The 9km Cycle 3 simulation uses active deep convection parametrization (Deep On).


**Figure D2: Zonal-mean precipitation for the individual months in the first year of Cycle 2 (dashed) and Cycle 3 (solid)**
**simulations.** IFS-NEMO 9km simulations are in blue, while IFS-FESOM simulations are given in red (4.4km), orange
(2.8km), and purple (9km).



**Appendix E**



**Figure E1: Mean changes in sea ice concentration (a) and significant wave height (b) between the 4.4km and 9km IFS-FESOM simulations in nextGEMS Cycle 3 (4.4km minus 9km). Red (blue) indicates an increase (decrease) for the 4.4km simulation.**



**Code Availability**

The FESOM2.5 model is a free software and available from Github (https://github.com/FESOM/fesom2). The latest version 2.5 including all developments used in nextGEMS Cycle 3 is archived in a Zenodo repository, https://doi.org/10.5281/zenodo.10225420 (Rackow et al. 2023c). The ocean coupling interface to the Integrated Forecasting System (IFS) has been extracted for IFS-FESOM and is publicly available as part of the FESOM2.5 code above as well (folder ifs_interface). MultIO, MIR, ECCODES and FDB are all free software and available at the ECMWF Github space, https://github.com/ecmwf. The IFS source code is available subject to a licence agreement with ECMWF. ECMWF member-state weather services and approved partners will be granted access. The IFS code without modules for data assimilation is also available for educational and academic purposes via an OpenIFS licence (see http://www.ecmwf.int/en/research/projects/openifs). For easier public access and review, the IFS code modifications from this study and developments detailed in section 3.1.1 for nextGEMS have also been separately archived in a Zenodo repository, https://doi.org/doi/10.5281/zenodo.10223576 (Rackow et al. 2023b). Scripts and data to reproduce the figures and analysis of this paper can be found at https://github.com/trackow/nextGEMS-paper/ (will be in Zenodo after review). Grib data in FDB were made available to hackathon participants using gribscan (Kölling, Kluft, and Rackow, 2024).

**Data Availability**

Data for our simulations are openly accessible and can be obtained either from the web (see DOIs below), from ECMWF's MARS archive, or directly from DKRZ's supercomputer Levante after registration (https://luv.dkrz.de/register/). The Cycle 2 data for 20 January 2020 to 31 December 2020 of TCo2559-NG5 with deep convection parametrization disabled can be found at https://dx.doi.org/10.21957/1n36-qg55. The Cycle 2 data for TCo1279-ORCA025 (20 Jan 2020 to 31 December 2021) with deep convection parametrization active can be found at https://dx.doi.org/10.21957/x4vb-3b40. More Cycle 2 output, also for the nextGEMS sister model ICON, can be found at the World Data Center for Climate (WDCC), https://dx.doi.org/10.26050/WDCC/nextGEMS_cyc2. Cycle 3 data for ICON and IFS can be found WDCC under https://doi.org/10.26050/WDCC/nextGEMS_cyc3 (Koldunov et al. 2023). Namelist files to reproduce the settings of the ocean, atmosphere, land, and wave model in the Cycle 3 simulations are archived in a Zenodo repository (Rackow et al. 2023a), https://doi.org/10.5281/zenodo.10221652. LSA SAF LST data are available from the LSA SAF data service under the link https://datalsasaf.lsasvcs.ipma.pt/PRODUCTS/MSG/MLST/.

**Author contributions**

TR led the writing of the paper and prepared the initial manuscript with TB and XPB. TR, TB, XPB, and IH performed the simulations. TB, XPB, RF, MD and TR developed the model code changes. The refactoring of the FESOM model has been led by DSi, NK, JS, and JH. Initial implementation of the IFS-FESOM single-executable coupling is joint work of KM and TR. NK created the 5km nextGEMS FESOM grid NG5 in discussions with TR. IP performed the QBO analysis. TB has analysed the precipitation characteristics and performed the TOA tuning. XPB contributed the 5-year temperature timeseries. SM performed TOA budget analyses. DT performed the MJO analyses. JB and JK contributed the wave model analyses. The city and urban heat island analyses are by XPB and ED. Sea ice performance indices are the work of LZ. TR performed the sea ice lead analysis. HFG provided ocean grid descriptions for coupling weight computations. In the paper, MD discussed the mass fixer approach and RF discussed the physics parametrizations. DSá added the multIO section to the paper. TK, LK, and FZ helped with faster data access. All co-authors discussed and contributed to the final document.



**Competing Interests**

The authors declare that they have no conflict of interest.

**Acknowledgements**

This work used supercomputing resources of the German Climate Computing Centre (Deutsches Klimarechenzentrum, DKRZ) granted by its Scientific Steering Committee (WLA) under project ID 1235. We want to thank DKRZ staff for their continued support in terms of data handling, data hosting, and running of the presented Cycle 3 simulations, in particular Jan Frederik Engels, Hendryk Bockelmann, Fabian Wachsmann, Irina Fast, and Carsten Beyer. We want to thank colleagues at AWI for active discussions and their support towards the upcoming multi-decadal simulations, in particular Suvarchal Kumar Cheedela, Bimochan Niraula, Rohit Ghosh, Sergey Danilov, and Patrick Scholz. We also would like to thank all colleagues at ECMWF who are not co-authors but also had a substantial impact on km-scale model development and modelling on climate timescales, e.g. Gabriele Arduini, Gianpaolo Balsamo, Magdalena Alonso Balmaseda, Margarita Choulga, Jasper Denissen, Charles Pelletier, Christian Kuehnlein, Pedro Maciel, Joe McNorton, Simon Smart, Balthasar Reuter, James Hawkes, Philipp Geier, Andreas Mueller, Michael Lange, Olivier Marsden, Sam Hatfield, Willem Deconinck, Matthew Griffith, Shannon Mason, and Mark Fielding. We thank Philippe Lopez for providing the Meteosat 8 observations in Figure 6. We also want to thank the international nextGEMS hackathon community, including many Early Career Researchers, who analysed our simulations in detail and helped guide some of the model development efforts. This research has been supported by the European Commission Horizon 2020 Framework Programme nextGEMS (grant no. 101003470). This work was also supported by the European Union's Destination Earth Initiative and relates to tasks entrusted by the European Union to the European Centre for Medium-Range Weather Forecasts implementing part of this Initiative with funding by the European Union.

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
