# Peer review of "Multi-year simulations at kilometre scale with the Integrated Forecasting System coupled to FESOM2.5/NEMOv3.4"

_EGUsphere, 2024_

## Author Comment (AC1)

**Response to reviewers**

We would like to express our sincere gratitude to the reviewers for their reviews. Following their request, we have revised the paper as follows. The referee comments are repeated in *italic blue* text, while our response is in black. During the revision process, work of other authors was included and we have updated the co-author list of the paper accordingly.

**Best regards, Thomas Rackow / On behalf of all co-authors**

**Anonymous Referee #1 (https://doi.org/10.5194/egusphere-2024-913-RC1):**

This is a report on an ECMWF contribution to the nextGEMS project consisting of a 4.4 km and 2.8 km version of IFS coupled to two eddy-permitting ocean models, NEMO and FESOM. Accurate coupled 5-year simulations, especially with IFS-FESOM, are a nice achievement on their own and a step toward the nextGEMS goal of multidecadal km-scale ocean-coupled simulations using seamless models that can skillfully span weather and climate timescales. A variety of improvements over ECMWF's operational IFS-NEMO model (9 km atm/0.25 degree ocean) are implemented, including global water and energy fixers, cloud tuning for radiation balance, a scale-aware parameterization of convective mass flux, ocean cooling from snow melt, Antarctic meltwater runoff, better land, urban and snow schemes, etc., in combination producing an impressively realistic climate simulation. Some phenomena that require km-scale to simulate are highlighted, including atmosphere-ocean interaction around leads, urban heat islands, and a QBO of realistic period driven primarily by resolved-scale gravity waves. Overall, the paper is well written and the results are state-of-the-art. I appreciated the illuminating discussion of the model development process, proceeding through three major iterations.

We thank the reviewer very much for this positive and encouraging summary. The paper indeed includes many important milestones at once that were achieved over the three development cycles of the nextGEMS project (and combining them with on-going developments from ECMWF, such as the new urban scheme). We sincerely hope that the experience gained here will be of help to the modelling community now that it is being documented openly.

It would have been nice to see a bit more analysis of the simulated ocean state, including mesoscale eddy statistics and vertical structure. The 2m temperature bias over ocean in the 5th year of the IFS/FESOM simulation looks remarkably small in Fig. 10, which is encouraging, but were other characteristics such as eddy kinetic energy, mean currents, thermocline depth, etc. examined? These characteristics do evolve on sub 5-year timescales and are relevant to the 30-year performance of a coupled model, so they seem in scope for this paper.

We agree with the reviewer that the ocean diagnostics were not the main focus in the previous manuscript, although especially in coupled models the ocean state is of fundamental importance. We have now performed a much more detailed analysis of the simulated ocean state, namely eddy variability and mean currents/sea surface height, as

well as ocean mixed layer characteristics in order to document some vertical characteristics as well. The analysis is given below and has been added to the new Section 3.2.3 of the paper.

Daily sea surface height (SSH) data is taken from the IFS-FESOM outputs and compared with the AVISO ("Archiving, Validation and Interpretation of Satellite Oceanographic data") multi-satellite altimeter data of daily gridded absolute dynamic topography, representing the observed SSH (Pujol et al. 2016). While ocean eddy variability in the 4.4km IFS-FESOM Cycle 3 simulation and AVISO was diagnosed from standard deviation of sea surface height, the structure of mean currents is diagnosed from the time-mean SSH.

Both the time-mean and variability of SSH show an excellent agreement between the simulation and observations from AVISO (new Fig. 11). The position of the main gyres and the gradient of SSH is well-reproduced, indicating a good performance in terms of position and strength of the main ocean currents. Ocean eddy variability is very similar as well with the eddy-resolving NG5 grid that has been introduced for IFS nextGEMS simulations (see Fig. B1 in Appendix B). However, while there are indications of it, the North Atlantic Current as northward extension of the Gulf Stream still underestimates SSH variability over the North-West corner. Moreover, Agulhas rings forming at the tip of South Africa seem to follow a too narrow, static path compared to observations.

---

## Author Response (AR1)

**Response to reviewers**

We would like to express our sincere gratitude to the reviewers for their reviews. Following their request, we have revised the paper as follows. The referee comments are repeated in *italic blue* text, while our response is in black. During the revision process, work of other authors was included and we have updated the co-author list of the paper accordingly.

Best regards, Thomas Rackow / *On behalf of all co-authors*

**Anonymous Referee #1 (https://doi.org/10.5194/egusphere-2024-913-RC1 ):**

*This is a report on an ECMWF contribution to the nextGEMS project consisting of a 4.4 km and 2.8 km version of IFS coupled to two eddy-permitting ocean models, NEMO and FESOM. Accurate coupled 5-year simulations, especially with IFS-FESOM, are a nice achievement on their own and a step toward the nextGEMS goal of multidecadal km-scale ocean-coupled simulations using seamless models that can skillfully span weather and climate timescales. A variety of improvements over ECMWF's operational IFS-NEMO model (9 km atm/0.25 degree ocean) are implemented, including global water and energy fixers, cloud tuning for radiation balance, a scale-aware parameterization of convective mass flux, ocean cooling from snow melt, Antarctic meltwater runoff, better land, urban and snow schemes, etc., in combination producing an impressively realistic climate simulation. Some phenomena that require km-scale to simulate are highlighted, including atmosphere-ocean interaction around leads, urban heat islands, and a QBO of realistic period driven primarily by resolved-scale gravity waves. Overall, the paper is well written and the results are state-of-the-art. I appreciated the illuminating discussion of the model development process, proceeding through three major iterations.*

We thank the reviewer very much for this positive and encouraging summary. The paper indeed includes many important milestones at once that were achieved over the three development cycles of the nextGEMS project (and combining them with on-going developments from ECMWF, such as the new urban scheme). We sincerely hope that the experience gained here will be of help to the modelling community now that it is being documented openly.

*It would have been nice to see a bit more analysis of the simulated ocean state, including mesoscale eddy statistics and vertical structure. The 2m temperature bias over ocean in the 5th year of the IFS/FESOM simulation looks remarkably small in Fig. 10, which is encouraging, but were other characteristics such as eddy kinetic energy, mean currents, thermocline depth, etc. examined? These characteristics do evolve on sub 5-year timescales and are relevant to the 30-year performance of a coupled model, so they seem in scope for this paper.*

We agree with the reviewer that the ocean diagnostics were not the main focus in the previous manuscript, although especially in coupled models the ocean state is of fundamental importance. We have now performed a much more detailed analysis of the simulated ocean state, namely eddy variability and mean currents/sea surface height, as

well as ocean mixed layer characteristics in order to document some vertical characteristics as well. The analysis is given below and has been added to the new Section 3.2.3 of the paper.

Daily sea surface height (SSH) data is taken from the IFS-FESOM outputs and compared with the AVISO ("Archiving, Validation and Interpretation of Satellite Oceanographic data") multi-satellite altimeter data of daily gridded absolute dynamic topography, representing the observed SSH (Pujol et al. 2016). While ocean eddy variability in the 4.4km IFS-FESOM Cycle 3 simulation and AVISO was diagnosed from standard deviation of sea surface height, the structure of mean currents is diagnosed from the time-mean SSH.

[revised manuscript text omitted]

*It is sometimes claimed that km-scale climate modeling is fundamentally simpler than 100 km grid climate modeling due to less reliance on poorly constrained parameterizations of subgrid variability, e.g. orographic gravity waves/drag, deep convection, subgrid cloud heterogeneity, etc.  The IFS experience seems to be that this is only partly true - the simulations are best if those subgrid parameterizations are still active at this scale, even though resolved-scale motions are doing more of the work.  This point might be made a bit more clearly in the conclusion to the paper.*

Thank you for this suggestion. We have added this point to the last paragraph of the paper, in the discussion section:

"At the scales used in this study, some modified subgrid parameterizations (e.g. deep convection with reduced cloud base mass flux) are still active for best performance, even though the influence of resolved-scale horizontal and vertical motions increases."

**Anonymous Referee #2 (https://doi.org/10.5194/egusphere-2024-913-RC2):**

*Introduction*

*The authors document the development of a new IFS-FESOM atmosphere-ocean coupled simulation, alongside discussion of the same atmosphere model coupled with the NEMO ocean model. The novel element of wide interest is discussion of the experiences of running this system at km-scale resolution for global domains over multi-annual period. This represents a substantial achievement, at the forefront of weather and climate model development, and therefore the work merits publication.*

We thank the reviewer very much for this positive evaluation of our manuscript.

*Given the significance of this development, however, I would recommend that the manuscript should be revised prior to acceptance. In the view of this reviewer, the resulting paper would have considerably more impact, if the following general remarks can be considered and addressed.*

We have followed many of the general remarks, as listed below. As the other reviewers have either confirmed that the length of the manuscript is reasonable (referee #3) or have asked us for additional analyses (referee #1), we could not follow all of them. Nevertheless, we think we have restructured and cut the paper in relevant parts, such as in section 2, in the abstract, or in lines 192-198 of the originally submitted manuscript, and thus made the study easier to follow and more concise, as suggested by the reviewer.

=====

*A) Paper length:*

*The manuscript represents a considerable development (as illustrated by substantial number of authors in the team), with multi-component codes brought together within a common modelling framework alongside supporting infrastructure. Even accounting for this, the paper is overly-long in my view, running to 59 pages in draft form with 16 main Figures and a further 5 Supplementary Figures within Appendices A-E. Whilst it is valuable to capture many of the details, a more rigorous review in general across all sections to assess where extraneous details are described might encourage wider readability. Some more specific suggestions are captured in remainder of this review, but I also suggest the authors should identify any opportunities to further streamline the text (including use of more summary Tables where relevant), reduce number of Figures etc.*

We thank the reviewer for this valid comment. The paper has been designed to give an overview about many different aspects that have been worked on by a substantial number of authors, and that is why it is difficult to cut the manuscript significantly without losing important aspects for other sections. Having said this, we streamlined the paper throughout many sections including the Introduction, and we cut independent paragraphs, such as the discussion of the RAPS environment (lines 192-198 in the original submitted manuscript) and the technical beginning of section 2.3 on the refactoring of the ocean model (l.223-229). In terms of the number of figures, to follow the reviewer's advice in some way we have decided to not include more Supplementary Figures that have been produced during the revision process. Those are just provided here in the response letter. An exception are the analyses asked for by other reviewers.

====

*B) Paper focus:*

*Given the range of simulation experiments conducted (Table 1), it is difficult to maintain a clear sense of the key research focus of the paper. For example, the abstract highlights a "focus mainly on….IFS-FESOM…with atmospheric resolution of 2.8km and 4.4km", while much of the analysis (e.g. Fig 5, 6, 9, 10, 11, 12, 13, 14, 15?, 16) seemed to focus more on comparisons between 4.4km and 9km systems, and at times more on comparisons between IFS-NEMO and IFS-FESOM. Greater clarity is therefore required as to the key focus for the documentation here. There is merit for GMD in focus on:*

1. *discussion of system innovations and improvements between Cycle1, Cycle2 and Cycle3 for a particular configuration,*
2. *discussion of the sensitivity of simulations to model resolution (across 9km, 4.4km and 2.8km if a common model framework can be used for this, or focus more deliberately on 9km vs 4.4km if simulation data better suited to this),*
3. *discussion of the impact of changing ocean model between FESOM and NEMO.*

*At present the paper attempts to capture certain aspects of all 3 approaches highlighted above, but this arguably makes for a relatively confused (and long) narrative for readers.*

The key focus on the 4.4km IFS-FESOM configuration in our study was now made clear in the abstract. This is the main target configuration and accordingly the 4.4km IFS-FESOM is also included in every single figure (an exception is Figure 6 where IFS-NEMO is used at 4.4km for technical reasons, see our comment below).

The 2.8km IFS-FESOM simulation is indeed only used in Figure 3, 4, 7, 8, so we have followed the reviewer's advice and made clear that this is only a supporting simulation.

We have also cut the abstract where technical details were given: *"This is enabled by a refactored ocean model code that allows for more efficient coupled simulations with IFS in a single-executable setup, employing hybrid parallelisation with MPI and OpenMP."* was removed.

====

*C) Paper structure:*

*Linked to comment B, the paper structure and sign-posting would merit a review and improvement by authors. Introduction of Table 1 (and Section 2.4 in general) earlier in the discussion of Section 2 might both help to reduce the overall length and make the discussion easier to follow.*

We thank the reviewer for this suggestion, as it makes it indeed easier to follow. We have now moved Section 2.4 up in the structure of Section 2. It is now the new Section 2.2, nextGEMS is introduced earlier, and the following sections are better understandable. We also added "...and Cycles" to Section 2.2 to make clear that information on those is found in this section. In addition, we have re-organized the headers and structure within Section 3 for increased readability. Thanks again!

*Section 2.1 is then rather exhaustive when not yet set in context of the nextGEMS runs, including perhaps some discussion on operational IFS which could be omitted where/if not directly related to the results presented (e.g. introduction of IFS-NEMO with ORCA12 as specific but not only example).*

This section introduces all necessary components of the IFS, such as the atmospheric component, the land component and wave model. NEMO being the standard ocean model

probably has to be explained here. As for FESOM, this is the first time that the model is introduced coupled to IFS, so the level of detail is unfortunately necessary here.

However, as suggested by the reviewer, we have cut an entire paragraph where the RAPS running/building environment is discussed, as this discussion is probably only necessary for scientists within the project and can be safely ignored without losing important aspects for the following sections.

*Given its importance to help orientate readers on the scope of the following discussion, I suggest improving Table 1 to include more details as easy reference, including some of the text from caption (e.g. translation of atmosphere resolution) into the main table, along with any other relevant details from the text.*

Done. We have changed the table layout and provide the atmospheric resolution here in a separate column. The ocean column was renamed accordingly to guide the reader that the ocean resolution info can also be found there.

*Section 3 highlights specific model developments within Cycles, alongside a discussion of the impact on results. This was not always straightforward to follow given the variety of topics discussed, so at least improved sign-posting and sub-sectioning would be useful (e.g. distinction between sub-parts of 3.1.1 and other discussion under 3.1.2+ not clear).*

The structure of Section 3 is that, following a short overview of identified key issues and developments at the beginning that some readers might want to skip, but which model developers reading GMD will find useful, we present how those successive development steps translate to a better representation of the coupled physical system in the following subsections. This might not always be the most natural structure, but we have followed it consistently for atmosphere and ocean/sea-ice components. We explain the chosen structure at the very beginning of Section 3.

*Results presented then begin to introduce elements and discussion of the comparisons across development Cycle as well as to model resolution and model components highlighted under my comment B). In general, I would suggest that Section 3 and thereby also Section 4 be reorganized to document more cleanly between these 3 aspects - or indeed be clearer in focus if only 1 or 2 of these considerations are to be discussed here.*

Please see our comments and argumentation below on the suggestion of reorganising Section 3 and Section 4 and why we have, with the exceptions listed below, stayed mostly with the original organisation. We trust that the reviewer will find these arguments compelling and aligned with the overall objectives of our study.

*Given concerns over paper length and clarity, I ask the authors to consider and provide a clear defence of the value of keeping discussion of Cycle 1 results within scope of this paper, in preference to a cleaner focus on Cycle 2 vs Cycle 3 developments here. Have Cycle1 to Cycle2 improvements been documented in other papers to date?*

There is no previous paper from the nextGEMS project that would detail the improvement from Cycle 1 to Cycle 2. The water and energy conservation in Cycle 1 versus Cycle 2 is

also key for ECMWF's improved operational numerical weather prediction (NWP) model and has not been documented in a paper. It is one of the first examples that prove that our seamless model development approach, where numerical weather prediction models are extended for km-scale multi-decadal climate applications, is a very useful approach to bridge the "Weather–Climate Schism", a term coined by Randall and Emanuel in their 2024 study. We show here that climate modelling activities can benefit the original NWP application as well and we think that encouraging examples in this direction merit publication in GMD.

*In general for all Figures from Fig. 3 onwards, a different blend of simulation results are plotted, but not always as a full consistent set, and thereby at times highlighting impact of Cycle changes, sometimes resolution, sometimes system differences, and sometimes a combination of some of these.*

These choices have occasionally been made to reduce the number of plotted lines, while still being able to make the point we wanted to make in the respective plot. As the reviewer notes further down as well, this seems to be somewhat desirable and matches the reviewer's suggestions, as we cannot overwhelm the reader with a consistent (large) set in all plots. We tried to use many simulations at once only where necessary, e.g. in Figure 7 (see also your comment under point D).

*Figures 6 and 7 for example, and accompanying discussion in 3.1.4 would sit better as a discussion of model characteristics for precipitation as a key variable of interest and sensitivity to treatment of convection at 9km and 4.4km than it being somewhat lost among other parts of system development discussion.*

The treatment of convection is an important part of our paper and the model results/characteristics for precipitation are closely related to our development/tuning efforts. Although we have a Section 4 with selected examples where the precipitation characteristics could arguably be presented instead, we think this discussion fits better closer to the model development discussion in Section 3 as there was a continuous back and forth between analysing new results, new ideas on how to better treat convection, followed by another round of results. The selected examples in Section 4 are different in this regard as they have not been the focus of our tuning and model development efforts but came out "naturally" with the move towards our final 4.4km configuration. We note, however, that all structures clearly have their advantages and disadvantages.

*In passing, I will note again that Figure 6 is focussed on 4.4km vs 9km IFS-NEMO (rather than 4.4km and 2.8km IFS-FESOM as per abstract stated focus?).*

As stated above, we have changed the sentence in the abstract as the "red thread" of our paper is clearly the discussion of 4.4km IFS-FESOM simulations (always plotted in "red" by the way), as also pointed out by the reviewer. We also made clear why IFS-NEMO appears here exceptionally (for technical reasons the simulated satellite images are only available with IFS-NEMO at the moment), see also our answer to your comment under point D. We have added this information to the caption of the figure where the simulated satellite image appears.

*By Figure 9, the results in Section 3 appear less concerned with impacts of Cycle developments, but back to discussion of comparative impact of ocean model and/or model resolutions. I suggest that with some reorganisation, the "selected examples" documented in Section 4 might then appear to be less standalone collection of somewhat ad-hoc topics, with sections 4.2 and 4.3 relatively weaker in terms of depth of analysis presented (and again reflecting some blend between a discussion of Cycle impacts and model resolution and codes, rather than being consistently focused on one specific strand).*

We thank the reviewer for the valuable feedback. While we understand that the selected examples in Section 4 might initially appear somewhat ad-hoc, this seems to be a matter of taste and we intentionally highlighted these diagnostics separately to ensure they are not lost among other discussions. As you noted in another part of your review, there is the danger that the novel scientific findings of our study could be lost between the many model developments we are describing, so this might be an understandable choice as the examples deserve focused attention.

We chose these specific highlights because they are novel and somewhat surprising, coming out "naturally" with the move towards our final 4.4km IFS-FESOM configuration in Cycle 3, and they cover a broad spectrum of the Earth System. They include aspects of land (cities), of the coupled atmosphere-ocean-sea ice system (sea ice leads), and atmosphere (MJO characteristics), thus providing a broader view of aspects of the model behaviour that have not been covered in many papers before. We believe Section 4 adds clear value to our paper. We hope the argumentation above clarifies our rationale a bit better.

*As highlighted by Reviewer 1, the lack of more detailed discussion of Ocean results is an omission, but given my concerns over paper length and focus, I might recommend being more deliberately focussed on only some aspects of atmosphere model performance here for example (i.e. Section 4.2 is rather 'random' in context of rest of the discussion), and encourage discussion of ocean and sea ice aspects to a second accompanying paper for example?.*

NextGEMS and the related Destination Earth project are fast-moving projects so that the ocean and sea ice aspects listed in the revised manuscript for Cycle 3 of nextGEMS are indeed an important baseline for the upcoming longer (30-year) simulations. It is likely that a second or third accompanying paper will not be written for this model cycle. We therefore wanted to make sure to include the basic ocean and sea ice discussion in this study that was asked by the other reviewer. As you will see above in our answer to reviewer #1, we followed reviewer #1's advice to add more ocean diagnostics but followed the second reviewer's advice to cut/revise the paper for other parts of the study.

*====*

*D) Figure quality:*

*In general the paper is well written, and Figures of a good quality for publication. However, I would also suggest to ensure:*

1. *Can any figures be provided with fewer simulations included (e.g. if discussion relates more specifically to impact of model Cycle, is this more clearly identifiable with fewer lines – see Figure 4 for example)*

We have given this a try but have not succeeded. For example, Figure 4 with only Cycle 3 data looks not much less busy as those simulations are all overlapping the observational grey area, but the information that Cycle 2 is still somewhat outside of the target range is completely lost then. We would therefore be tempted to include another Supplementary Figure for Cycle 2 exclusively, which we feel is however not helping the overall length of the paper.

As the other reviewers have not voiced a similar concern, we hope that the reviewer will find the final high-res figure (in PDF format) to be clearly distinguishable and therefore accessible for everyone.

*Aim for more consistent inclusion/focus on certain experiments (e.g. choice of IFS-NEMO in Fig 6 rather than IFS-FESOM seemed somewhat arbitrary?).*

This is a very understandable comment, and we have also struggled with this choice at first. While the choice seems arbitrary, we were forced to use IFS-NEMO here as the simulated satellite image on the left side of the plot is, for technical reasons, just readily available in the workflow that the operational IFS model is using, for which only IFS-NEMO is available at the moment. We have clarified this in the caption of the figure. Given the short length of this test simulation, we can assure you that IFS-NEMO and IFS-FESOM look similar here.

2. *Figure 7 – this is an important plot in context of km-scale global model development, but the detail is a bit lost with smaller sub-panels and number of lines. What is the key sensitivity to be drawn out here? 6 model experiments are listed in the legend for example, but I'm not sure I can differentiate more than 5 lines (maybe Cycle 3 9km IFS-NEMO and IFS-FESOM overlapping??).*

Yes, in the left plot two lines are surprisingly close and overlapping (Cycle 3, 9km, IFS-FESOM and IFS-NEMO). You can zoom in at around $10^{-1}$ mm/hour and see a tiny bit of difference but otherwise the lines are on top of each other. This means the influence of the ocean model on precipitation intensity is very small, at least when not running the coupled model for more than 1 year. In the right plot which gives some information about system sizes and thus informs about convective organisation you see all 6 lines individually. We want to point out that all solid lines are for Cycle 3, all dashed lines are from Cycle 2, so that one can see the impact that model developments towards Cycle 3 brought with them. Also, red colour hues are for the IFS-FESOM runs, and blue is reserved for reference runs with IFS-NEMO. The purple line mixes the coarser atmospheric resolution of 9km (usually in "blue" IFS-NEMO runs) with the use of FESOM (usually "red").

The key sensitivity to be drawn out of here is that by adapting the convection scheme by reducing the cloud base mass flux, the precipitation characteristics change substantially, with more realistic intense precipitation when the new settings are used (Cycle 3, 4.4 km), while otherwise with deep convection switched off (Cycle 2, 4.4 km and 2.8 km) intense

precipitation is overestimated, and in the 9km simulations with deep convection scheme fully active, intense precipitation is underestimated.

*Figure 8 – similar comment on difficulty capturing results beyond "all in the bunch". Suggest plotting as differences relative to GPM rather than absolute zonal average precipitation may be preferable here to focus on sensitivities.*

We understand the difficulty. However, just to show the difference relative to GPM is not an option (we have tried) because the information of where the maxima in precipitation are located is necessary in order to understand the plot. The only thing we could do is to add the proposed figure to the Supplementary Material, but for now we kept it. As the figure will be relatively high-res in the final PDF format after publication, we think one can see all important aspects when zooming in.

*Keeping consistency in sub-panel layout will help readability (e.g. a = IFS 9km, b = IFS 4.4km in Fig 9, but opposite way round by Fig 10). Similarly, keeping consistency of model framework – Fig 9 and Fig 10 both representing comparison of results with both changed resolution and changed ocean model – is it ocean model or atmosphere resolution that dictates here, and if known therefore present the direct comparison.*

Thank you for spotting this inconsistency in the layout. We have updated the QBO plot and its caption with a more consistent labeling that is used throughout the rest of the paper. We have also used this opportunity to extend the ERA5 timeseries in this comparison to the end of August 2024.

Moreover, we updated figures 13 and 14 (now 15 and 16) with more consistent titles.

[Figure]

3. *Figure 15 – I personally found the 'hatching' across land areas more confusing/distracting than helpful on first reading. Combining this discussion more directly with/into 3.2.3 would help make the paper less disjointed.*

Although we have not moved the discussion within the paper, we have redone this plot as suggested by the reviewer and removed the distracting hatching. We are now using a single contour instead to show the border between land and ocean areas now, as follows:

[Figure]

We hope this will make the figure easier to understand.

*Finally, Figure 16 provides a cleaner comparison of impact of Cycle change with common model resolution and framework (so arguably more of a 'Section 3' item in the current paper structure?). Are the authors able to clarify (isolate) whether the differences shown are a function of the change in LULC, or the addition of the urban scheme, or some useful combination of both changes? Given the substantial change between panels c) and d), it is tempting to think it is mostly the LULC update that drives the improvement?*

The reviewer raises an interesting question in trying to attribute the surface improvements to specific model developments. McNorton et al. (2023) already showed the isolated impact of activating the urban scheme on shorter timescales. To properly answer the reviewer question, however, we would ideally need an equivalent experiment to Cycle 3 with only one of the two modifications: either new LU/LC maps, or the urban scheme, which we do not have at the moment due to enormous computational costs of each of these runs.

However, we have hints pointing to the fact that both contributions can lead to an improved performance individually, with a city-dependent relevance of each development. For Warsaw, an analysis on the skin-temperature mean diurnal cycle of the rural and urban areas separately suggests that improvement from Cycle 2 to Cycle 3 come mostly from a better representation of the *rural* areas. However, for other cities, e.g., London, the improvement comes mostly from the *urban* areas. Note that LSA-SAF observations are likely to have significant biases in their measured absolute temperature. However, looking at the rural-urban contrast removes a large part of such bias.

McNorton, J., Agustí-Panareda, A., Arduini, G., Balsamo, G., Bousserez, N., Boussetta, S., et al. (2023). An urban scheme for the ECMWF Integrated forecasting system: Global forecasts and residential $CO_2$ emissions. *Journal of Advances in Modeling Earth Systems*, 15, e2022MS003286. https://doi.org/10.1029/2022MS003286

[Figure]

[Figure]

=====

*Summary*

*In summary, this is a substantial and noteworthy paper documenting considerable technical and scientific progress. However, in its current form, I do not think the paper is reporting that achievement as well as it could, and hope the reflection above highlights some areas that might support further improvement prior to publication.*

*I would be happy to provide further review if useful, including potential for more specific review where a shorter and more focussed paper might make this more tractable.*

Thank you again for the overall very positive assessment. We have followed some of the general remarks in terms of presentation, with regards to structure, length, and figure layout as listed above, while also trying to follow the other reviewers' remarks, and are confident that this has resulted in a good compromise and a better readable manuscript.

**Anonymous Referee #3 (https://doi.org/10.5194/egusphere-2024-913-RC3 ):**

*This manuscript provides an overview of multi-year km-scale global coupled simulations using the IFS coupled to two different ocean-sea ice models. In addition to a description of the various model components, the manuscript highlights the existing issues in the previous version of the model and the major model developments that address those issues for the new version of the model. In addition to highlighting the improvements in this version of the model, the authors also point out existing biases, which are just as valuable to document and share with the community in this burgeoning area of km-scale coupled simulations. The authors conclude the manuscript by providing the context in which the development of these models exist and how such high resolution climate simulations open the avenue for*

We thank the reviewer very much for this summary and share the opinion that remaining biases need to be pointed out and shared with the community in order to have a chance to solve them.

*These 5-year coupled simulations with two different ocean-sea- ice models at these resolutions is a considerable accomplishment, and a manuscript that documents these simulation results well warrants publication. The manuscript is clearly organized and written. The scope of this manuscript is large, covering the atmosphere, land, and ocean-sea ice models and that leads to quite a long manuscript, but considering that it is an overview paper, I believe the length is warranted. I only have the following minor comments that I believe will help provide more context to the baseline Cycle 3 simulations that are presented in this manuscript.*

We thank the reviewer for the positive assessment. It has indeed been a delicate balance between deciding how much material to show and still having a reasonable manuscript length, so we are happy that we seem to have found a reasonable compromise. The response to the minor comments is given below.

*Minor comments*

**More details about radiative fluxes in the Cycle 3 experiments**

*Based on Section 3.1.1, one of the key issues fixed in Cycle 3 appears to be the radiation balance of the model. While the Figure 4 is helpful in seeing the time evolution of the global mean top of atmosphere radiation bias with respect to CERES, it doesn't give us a sense of the partitioning of longwave and shortwave and geographic distribution. It is mentioned in Section 3.1.1 that changes in both low-clouds and high-clouds were made to reach a more balanced radiative balance. A geographic map of the change in the radiative fluxes would be useful in seeing how those change between Cycle 2 and 3 manifest themselves geographically and in the shortwave and longwave fluxes.*

Figure 4 shows the global mean top of atmosphere radiation bias with respect to CERES, as well as the longwave and shortwave radiation bias with respect to CERES (see two top panels on the left). We did not discuss geographic maps of the biases in the paper, as year-to-year variability can be large and patterns uncertain, and because the paper was already too long. To inform the reviewer, we have added maps of annual TOA radiation bias for the 4.4 km simulations in Cycle 2 and 3 below. The Figure shows that the main (and persistent) regional biases are positive shortwave biases due to underestimated stratocumulus clouds, located in the Pacific off the coast of North and South America and in the Atlantic off the coast of Angola and Namibia. There is also a persistent positive shortwave bias over China, presumably because the aerosol climatology is biased in this region. Positive shortwave biases in other regions like over the Maritime Continent and the Pacific ITCZ are to a large degree compensated by negative longwave biases in the same regions, indicating an underestimation of deep convective clouds in these regions.

[Figure]

annual TOA bias, IFS (2020) rel. to CERES climatology (2001-2021)

[Figure]

Compared to the persistent local biases, changes from Cycle 2 to 3 are relatively small, with the most significant reduction in shortwave and longwave bias over the Maritime Continent, where reducing the threshold that limits the minimum size of ice effective radius had a big impact. We have added a short paragraph to Section 3.1.3 that discusses the above points. Due to the length of the paper and comments by reviewer #2 we have decided against including the Figure in the manuscript.

***Thermodynamic and dynamic impact of the model developments***

*I would have expected that the fixes to the water and energy imbalance and changes to the deep convective scheme to have an impact on the thermodynamic (temperature and humidity) and dynamic structures within the atmosphere. Were they not reported here because the impacts were negligible? If so, a sentence or two describing the lack of impact would be informative to show that indirect impacts are small compared to the direct impact that are reported in this manuscript. If it did have an impact, I believe it would also be useful to show those changes, because this is an overview of these simulations and the changes in*

*other large-scale features from the changes are already reported (radiative fluxes and zonal precipitation).*

On climate time scales, the impact that the fixes to water and energy imbalance have on the temperature and humidity profile are small, and are dominated by the effects that correcting the energy leakage has. However, on time scales of numerical weather prediction, there are some noteworthy impacts on the thermodynamic structure of the atmosphere, as the mass fixers for all moist species slightly cool and dry the troposphere, and the reduced temperature biases were part of the reason why tracer mass fixers improved the skill scores of the operational IFS model and were part of the operational IFS upgrade in June 2023 (48r1). This was already mentioned in the initially submitted manuscript in Section 3.1.2 and in the "Summary and Conclusions". We have added a short discussion of the points above to Section 3.1.2 .

Deactivating the convection scheme for deep convection or using the setup with reduced cloud base mass flux (as in Cycle 3) has a substantial impact on the vertical temperature and humidity profile, particularly in the tropics. Due to length restrictions we do not want to discuss this in too much depth in this paper, but a separate paper is in preparation that discusses all nextGEMS developments that concern the convection scheme and other moist physics parameterizations in more detail, led by Tobias Becker. In the revised document, we now very briefly mention that with Deep Convection Off (Cycle 2 at 4.4 km), the tropical troposphere is biased warm and dry (Section 3.1.1), while with reduced cloud base mass flux (Cycle 3 at 4.4 km), the tropical troposphere is colder and more humid (Section 3.1.4).

*Specific (minor) comments*

*L394: qv, ql and qi are listed but they are referred to as water vapour, cloud ice and snow. Based on the equation, it should be qv, qi and qs.*

Done!

*L402: typo with Wm--2*

Done! This seems to have been caused by a conversion from Google doc to Word docx.

*L461-463: Do the authors have any more information of whether these weakly active convection events are deep events with little mass flux or whether they are shallow events?*

The weakly active convective events that cause a peak in the precipitation PDF between 0.1 and 1 mm/hour in the Cycle 3 simulation at 4.4km resolution are related to weakly active deep convection. Compared to the Cycle 2 simulation at 4.4 km resolution, the deep convection scheme has been reactivated (with a reduced cloud base mass flux). The shallow convection scheme remained unchanged between Cycle 2 and 3, so it is the deep convection scheme that causes the differences between the Cycle 2 and 3 simulations, and thus also the peak in the precipitation PDF between 0.1 and 1 mm/hour that can be seen in Cycle 3. However, note that compared to resolved deep convection, the deep convection

produced by the deep convection scheme reaches on average significantly less high. In the manuscript we added the information that the deep convection scheme is responsible for the peak between 0.1 and 1 mm/hour.

*L762-L764: I realise it might be difficult to explain, but do the authors have a hypothesis for why at higher resolution the eastward inertia-gravity waves and mixed Rossby-gravity waves might be better represented?*

What makes the interpretation of this point difficult is that there is not just a difference in spatial resolution between 9 km and 4.4 km simulations, but also in the treatment of deep convection. One of our speculations is that a reduced cloud base mass flux can improve the representation of the covariance between the upper-level heating and warm temperature anomalies because of more resolved moisture transport, which can amplify synoptic-scale wave activities. Nevertheless, other possibilities such as the impacts of the mean convective activities and circulations should be examined.